# Stability of Erythrocyte-Derived Nanovesicles Assessed by Light Scattering and Electron Microscopy

**DOI:** 10.3390/ijms222312772

**Published:** 2021-11-25

**Authors:** Darja Božič, Matej Hočevar, Matic Kisovec, Manca Pajnič, Ljubiša Pađen, Marko Jeran, Apolonija Bedina Zavec, Marjetka Podobnik, Ksenija Kogej, Aleš Iglič, Veronika Kralj-Iglič

**Affiliations:** 1Laboratory of Clinical Biophysics, Faculty of Health Sciences, University of Ljubljana, SI-1000 Ljubljana, Slovenia; darja.bozic@zf.uni-lj.si (D.B.); manca.pajnic@zf.uni-lj.si (M.P.); ljubisa.paden@zf.uni-lj.si (L.P.); marko.jeran@fe.uni-lj.si (M.J.); 2Department of Physics and Chemistry of Materials, Institute of Metals and Technology, SI-1000 Ljubljana, Slovenia; matej.hocevar@imt.si; 3Department of Molecular Biology and Nanobiotechnology, National Institute of Chemistry, SI-1000 Ljubljana, Slovenia; matic.kisovec@ki.si (M.K.); polona.bedina@ki.si (A.B.Z.); marjetka.podobnik@ki.si (M.P.); 4Laboratory of Physics, Faculty of Electrical Engineering, University of Ljubljana, SI-1000 Ljubljana, Slovenia; ales.iglic@fe.uni-lj.si; 5Faculty of Chemistry and Chemical Technology, University of Ljubljana, SI-1000 Ljubljana, Slovenia; ksenija.kogej@fkkt.uni-lj.si; 6Faculty of Medicine, University of Ljubljana, SI-1000 Ljubljana, Slovenia

**Keywords:** light scattering, vesicle characterization, vesicle stability, extracellular vesicle reference material, nanovesicles, cellular vesicles, vesicle shape, scanning electron microscopy of extracellular vesicles, cryo-electron microscopy of extracellular vesicles

## Abstract

Extracellular vesicles (EVs) are gaining increasing amounts of attention due to their potential use in diagnostics and therapy, but the poor reproducibility of the studies that have been conducted on these structures hinders their breakthrough into routine practice. We believe that a better understanding of EVs stability and methods to control their integrity are the key to resolving this issue. In this work, erythrocyte EVs (hbEVs) were isolated by centrifugation from suspensions of human erythrocytes that had been aged in vitro. The isolate was characterised by scanning (SEM) and cryo-transmission electron microscopy (cryo-TEM), flow cytometry (FCM), dynamic/static light scattering (LS), protein electrophoresis, and UV-V spectrometry. The hbEVs were exposed to various conditions (pH (4–10), osmolarity (50–1000 mOsm/L), temperature (15–60 °C), and surfactant Triton X-100 (10–500 μM)). Their stability was evaluated by LS by considering the hydrodynamic radius (*R*_h_), intensity of scattered light (*I*), and the shape parameter (*ρ*). The morphology of the hbEVs that had been stored in phosphate-buffered saline with citrate (PBS–citrate) at 4 °C remained consistent for more than 6 months. A change in the media properties (50–1000 mOsm/L, pH 4–10) had no significant effect on the *R*_h_ (=100–130 nm). At pH values below 6 and above 8, at temperatures above 45 °C, and in the presence of Triton X-100, hbEVs degradation was indicated by a decrease in *I* of more than 20%. Due to the simple preparation, homogeneous morphology, and stability of hbEVs under a wide range of conditions, they are considered to be a suitable option for EV reference material.

## 1. Introduction

Extracellular vesicles (EVs) are a heterogeneous group of nano- to micro-sized membrane-enclosed particles that are found in almost every sample of biological origin [1]. They function as a means of intercellular communication [1,2,3,4] and are involved in several physiological and pathological contexts, such as in embryogenesis [5,6,7,8,9], neuronal communication [10], blood coagulation [11,12], inflammation [13,14], tumorigenesis [1,15], and horizontal gene transfer [7,8,16]. In recent decades, EVs have been extensively studied for their potential clinical utility: as biomarkers to track the progression of various diseases, as drugs, or as vectors for drug-delivery [4,17,18]. However, the breakthrough from potential to practical application has been hindered by technical difficulties [19,20,21,22]. Due to the small size of EVs and their heterogeneity in terms of size and composition, the quantification of EVs is still a challenge [23,24]. In addition, the properties of the isolated material strongly depend on the method of isolation [25,26,27,28]. Even when the same type of equipment and protocols are used, the reproducibility of isolation tends to be poor [22,29].

To improve the comparability of studies and to avoid misconceptions in data interpretation, general guidelines [30] have been formulated by the International Society for Extracellular Vesicles (ISEV), according to which a comprehensive assessment using several different techniques is required to evaluate the size, concentration, topology, and phenotype of the particles in the samples. However, such an approach is very expensive and time consuming and therefore cannot be performed on a regular basis or with a high sample throughput. Although this is crucial for the effective and safe use of EV preparations and their mimics, the decision on how to monitor the integrity of the samples is not straightforward. Methods should be improved so that more information can be obtained from a single experiment. Direct approaches are desirable to keep the samples intact for further purposes, to reduce the cost and complexity of analysis, and to minimize the environmental footprint.

In order to compare the results of the EV analyses performed in different laboratories, the need to standardize analysis methods has been addressed [31], and recommended guidelines have been introduced by the International Society for Advancement of Cytometry (ISAC), the International Society for Thrombosis and Haemostatis (ISTH), and ISEV [32]. Erythrocyte vesicles have been suggested to be a possible suitable reference material by Valkonen et al. [33]. Recombinant vesicles are now available [34], but similar to other standards, they may not be accessible to every laboratory due to their relatively high price.

To improve the reproducibility of experiments, the better sample processing monitoring is crucial. The control over EV preparations is only possible by understanding the behaviour of the (current) sample under the conditions that are encountered during isolation, storage, and analysis. The stability of the cells (mechanisms of cell-directed and spontaneous vesiculation), EVs (resistance/dynamics of EV assembly), and other components in the sample must be considered.

In the past, the stability of EV-samples was mainly considered from the point of view of sample storage. As highlighted in a recent review by Yuan et al. [35], there is still a lack of knowledge on how to preserve samples or isolated EVs before the functional analysis or therapeutic application, and the results of the few studies that have investigated different temperatures and time periods of EV storage are inconsistent. Even less is known about the effects of other parameters, such as medium change. The aim of the present article is to make a contribution to aid in the filling of this gap.

Vesicles are a self-assembly structure that is shaped by non-covalent interactions. The pH and ionic strength of the medium affect the charge of the biomolecules in the sample and therefore modulate the nature and strength of the interactions between them. Temperature is another factor that affects the equilibrium of the system and causes reversible or irreversible changes. Studies on artificial phospholipid vesicles have shown that pH can alter the affinity between the phospholipids and cholesterol [36], which may influence the fluidity and phase separation of the membrane. It can also affect membrane permeability [37,38] and the activity of transmembrane transporters such as ATP-ase, modifying the flow of the ions and fluid across the membrane reflected in a change of vesicle size [39]. A change in the net charge of vesicles can trigger their aggregation [40]. The aggregation and eventual fusion of vesicles that are composed of lipids with negatively charged head groups, such as phosphatidylserine or phosphatidylethanolamine, can be triggered by H^+^ [41]. The tonicity of the medium can affect vesicle morphology [42,43] and membrane permeability [43]. It has been observed that fusion of vesicles can be promoted by the presence of divalent cations [41,44], a rise in temperature [44,45,46], or a high osmotic pressure gradient across the membrane [44]. The fusion events also appear to depend on the size [44] and concentration [46] of the vesicles. Some phospholipid vesicles have been found to aggregate reversibly upon cooling [47].

It was observed that incubation at low (pH = 4) or high pH (pH = 10) and at high temperatures (60 °C) or at freezing temperatures (at −20 °C and −80 °C) enhanced EV degradation and increased their uptake into the recipient cells [48]. To the best of our knowledge, a detailed study on the structural changes of natural EVs upon changes in pH, osmolarity, and temperature has not yet been performed.

For our study, erythrocytes were chosen as the source of EVs because they represent a well-studied “simple” cell model that allows a considerable amount of EVs to be harvested (a visible EV pellet was obtained from 5 mL of blood, which is sufficient to perform the presented set of experiments). Mature mammalian erythrocytes have no organelles and no internal scaffold and are therefore also a suitable model that can be used to study the very basic mechanisms of vesiculation. Their morphology is determined by the elastic properties of the erythrocyte membrane [49,50,51], including the contribution of shear stress deriving from the membrane skeleton [52,53] and the volume to area ratio given by the relative volume *v* = (*V*^2^/36 π*A*^3^)^1/2^, where *A* is the membrane area and where *V* is the enclosed volume. The relative volume is attained according to the equality of osmotic pressure inside of and outside of the cell [49,54] and to the electro-neutrality of both compartments. Recently, additional mechanisms that have been determined to be important for the shape of the erythrocyte, such as the orientational ordering of the membrane constituents, have been suggested [55,56]. The loss of the structural integrity of the membrane due to the storage lesion affects the deformability of the cells and their ability to recover from mechanical deformation, which is reflected in the quality and viability of the cells [57]. Upon in vitro storage, haemolysis can occur through the echinocyte (triggered by ATP deficiency) or stomatocyte (induced by the lowering of pH) transformation [58]. To form EVs, erythrocytes usually first undergo a discocyte–echinocyte transformation and present spicules. Budding occurs at the tip of the spicules. The membrane skeleton detaches from the lipid bilayer and does not enter the bud [59,60,61]. The pinched-off vesicles, which are usually about 100–300 nm in size, contain haemoglobin (hence we use the abbreviation hbEVs), acetylcholinesterase, band 3 protein, glycophorin A, and actin but are essentially free of spectrin and ankyrin [62,63].

hbEVs have been studied in the context of the erythrocyte life cycle, membrane properties, and storage lesions (reviewed in [64,65]). The latter is of particular interest because of the procoagulant properties of the vesicles that are formed in stored blood units [66,67], which may affect transfusion safety. It has been noted that the morphology and composition of the shed vesicles can change during the course of erythrocyte storage [63,68]. Several mechanisms for erythrocyte vesiculation have been proposed, with key events being attributed to the increase in the intracellular Ca^2+^ concentration that is associated with the loss of membrane asymmetry (erythrocyte apoptosis, [69]), the detachment of the cytoskeleton from the membrane [70], and protein oxidation followed by band 3 clustering and the subsequent aggregation of membrane and membrane-associated proteins [63,71]. Erythrocyte vesiculation can also be induced by surfactants [72,73].

In our study, the hbEVs were isolated from erythrocyte suspensions that had been aged in vitro. The samples were analysed by scanning electron microscopy (SEM), cryogenic transmission electron microscopy (cryo-TEM), flow cytometry (FCM), dynamic (DLS) and static (SLS) light scattering (LS), UV-Vis spectroscopy, and protein electrophoresis (SDS-PAGE). The hydrodynamic radius (*R*_h_) and intensity of the scattered light (*I*) were monitored to evaluate the stability of the hbEVs. Their cargo was evidenced directly by electron microscopy, and indirectly by the shape parameter (*ρ*), which was derived from the *R*_h_ (a DLS size parameter) and the radius of gyration *R*_g_ (a SLS size parameter) so that *ρ* = *R*_g_/*R*_h,0_, where *R*_h,0_ is the value of the *R*_h_ extrapolated to angle 0 (described in Methods). We examined the changes in the hbEV samples upon storage, the variation of osmolarity and the pH of the suspensions, and heating. To test the sensitivity of our DLS/SLS approach to detect the loss of sample integrity, we also examined the lysis of hbEVs with the surfactant Triton X-100. Triton X-100 was chosen because it is one of the most routinely used agents for the permeabilization of biological membranes.

## 2. Results

### 2.1. Mechanism of hbEVs Formation and Their Morphology

After a few days of storage in PBS–citrate at 4 °C, the majority of erythrocytes transformed into echinocytes, as observed by light microscopy (not shown). The budding of the erythrocyte membrane can be seen on representative SEM images of an echinocyte (Figure 1a). The hbEVs were isolated by a differential centrifugation protocol (Methods) from an erythrocyte suspension that had been stored at 4 °C for 6 days (SEM images of the pellets in selected steps of the isolation are provided in Appendix A). After the final isolation step, the red-coloured pellet was visible to the eye.

Cryo-TEM images (Figure 1b and Figure A2 and Figure A3 in Appendix B) confirmed that the isolate mainly contained bilayer membrane-enclosed vesicles with darker contents, indicating that the hbEVs were filled. Spectrometric assessment and protein electrophoresis (Appendix C) proved the presence of haemoglobin, and it was therefore concluded that the observed vesicles were filled with haemoglobin. According to SEM analysis, the hbEVs that were suspended in 300 mOsm/L PBS–citrate were globular, slightly elongated (Figure 1c,d), and corresponded to the size and shape of the echinocyte buds. Some tubular vesicles were also present in the isolates (Figure 1d and Appendix A). The reduction of the medium osmolarity induced a change in the average morphology of the hbEVs from elongated to more spherical (Figure 1e,f). Fewer and smaller vesicles were observed in the sample in the presence of Triton X-100 (150 μmol/L) (Figure 1g,h).

Representative FCM scatterplots of an hbEV sample that has been suspended in isotonic (300 mOsm/L PBS–citrate) or in hypotonic medium (150 or 50 mOsm/L PBS–citrate) are presented in Figure 2. Standard beads (0.5 μm, 1 μm, 2 μm, and 3 μm; analysed in the same FCM settings) are shown in panels (a) and (b) for comparison. In the FCM scatterplots, the hbEVs appear as a well-defined cloud of events, which was centred (in both of the axes) below the cloud corresponding to the 0.5 μm polystyrene beads (Figure 2a–c).

Over 20% more events were detected by FCM when the sample was suspended in hypotonic medium (50 mOsm/L PBS–citrate) than when the sample was suspended in the isosmotic medium, and a shift of the hbEVs event cloud to higher forward scatter (FSC) values with decreasing medium osmolarity was noted (Figure 2a–c). In an analysis of the serially diluted hbEV samples (Appendix D), it was observed that the density of the number of events that was detected by FCM was affected by sample concentration. Disproportionally more events per volume were measured in more concentrated samples, suggesting that co-events were detected. Further sample dilution did not resolve this issue (the concentration range resulting in detection of 200–10,000 events/s was tested), indicating that the hbEVs concentration was progressively underestimated at a higher dilution. It was concluded that due to their small size, only a small proportion of vesicles was detected by FCM at the applied instrument settings.

The size distributions of the hbEVs in different media with respect to *R*_h_ obtained by DLS analysis at 90° were monomodal, with the peak posed at *R*_h,90_ being between 100 and 150 nm (representative examples are shown in Figure 2g). The differences in the mean *R*_h,90_ and distribution widths pertaining to samples with different osmolarities were within the expected 10% error range (usually accepted for the DLS method), but there was a slight deformation in the peak towards a lower *R*_h,90_ in hypertonic medium (1000 mOsm/L) and towards a larger *R*_h,90_ in hypotonic medium (50 mOsm/L) indicated (Figure 2g), which is in line with the shift of events in the scatterplots that were observed by FCM and the morphological changes observed by SEM. DLS/SLS analysis showed similar *R*_h,90_ and *I*_90_ in the sample suspended in 50 mOsm/L PBS–citrate and in the sample suspended in 300 mOsm/L PBS–citrate, suggesting that the transfer of hbEVs to the hypotonic medium did not substantially change hbEVs size nor concentration.

Due to the evident dependence of the number of events that was detected by FCM on the osmolarity of the medium and due to disproportion between the number of events and the sample concentration, we concluded that the concentration values of the hbEVs that were determined by the FCM did not reach sufficient reliability. Therefore, stability analysis was performed using DLS/SLS.

### 2.2. Stability of HbEVs with Respect to Osmolarity of the Suspension, pH, Temperature and Addition of Surfactant Triton X-100, Determined by LS

The stability of the hbEVs was further assessed in terms of the effects of osmolarity, pH change, temperature, and the addition of the surfactant Triton X-100, using a single-angle (to detect possible size and concentration changes) and multi-angle DLS/SLS approach (to inspect possible structural changes). Figure 3 shows the changes of the mean *R*_h,90_ upon the change of osmolarity, the pH values of the media, temperature, and the addition of surfactant Triton X-100. The scattered light intensity at the angle 90° (*I*_90_) is reported with respect to the initial sample of hbEVs (hbEVs sample suspended in 300 mOsm/L PBS–citrate, pH 7.2, in the absence of Triton X-100, measured at 25 °C). At temperatures above 50 °C and at concentrations of Triton X-100 above 200 μmol/L, the absolute intensity of the scattered light *I*_90_ was too low to allow reliable *R*_h,90_ determination (measurements were highly variable), which is therefore not reported in Figure 3 (panels c,d). The correlation functions can be found in Figure A6, Appendix E.

It can be seen that *R*_h,90_ remained constant within the 10% error rate of the method in the tested osmolarity range (50–1000 mOsm/L), pH (= 4–10), and temperature range (15–60 °C) (Figure 3a–c**,** respectively), indicating that morphology of the hbEVs was largely preserved within these fairly wide intervals. A trend of a decreasing *R*_h,90_ (Figure 2g, Figure 3a,e, and Appendix E, Figure A6) with increasing osmolarity was observed and was accompanied by a moderate decrease in *I*_90_ (Figure 3a). Light scattering intensity increases with particle size according to the power law (e.g., as *R*^6^ for solid spherical particles); therefore, the observed trend was interpreted as the result of a change in the scattering properties of the particles rather than their degradation. While *I*_90_ remained above 90% of the initial value (pertaining to 25 °C) between 15–45 °C, it dropped significantly at higher temperatures, reaching almost 0 at temperatures above 55 °C. With respect to the conservation of *R*_h,90_, this steep drop in *I*_90_ was interpreted as representing extensive hbEVs degradation.

While *R*_h,90_ was preserved for increasing concentrations of added Triton X-100 throughout the concentration interval below 200 μmol/L, *I*_90_ gradually decreased in this interval and eventually reached 0 (Figure 3d). Vanishing *I*_90_ at higher concentrations of Triton X-100 indicates the degradation of hbEVs. However, multiangle analysis showed that slightly lower *R*_h_ values were obtained at the Triton-X100 concentration of 150 μmol/L than they were in the sample without the added surfactant (Figure 3e).

As little difference was observed in the single-angle experiments, multiangle assessment was performed on selected samples to expose possible subtler changes (Figure 3e–g, Table 1). Comparing the hbEVs suspended in an isotonic medium (at 300 mOsm/L) with the corresponding sample suspended in hypotonic medium (at 50 mOsm/L), we found an agreement in the angular dependencies of the two samples (Figure 3e–g). Treatment with the surfactant Triton X-100 at the concentrations of 100 μmol/L and 150 μmol/L did not cause large deviations of the scattering properties of the particles (Figure 3e–g), except for the significant drop in *I* at higher Triton X-100 concentrations, suggesting the degradation of a portion of the particles. As can be seen from the Kratky plot in Figure 3g, the measured data agreed well with the theoretical Guinier function, which was herein used to derive *R*_g_ of the hbEVs, in all cases. For all the investigated samples, the value of the shape parameter *ρ* was close to 0.8 (Table 1), which is characteristic of the topology of a “filled” sphere [74]; however, it was slightly higher (*ρ* = 0.81) for the sample that had been treated with Triton X-100 (Table 1). The detection of a slight elongation of the shapes in 300 mOsm/L PBS–citrate (observed by SEM, Figure 1) was beyond the sensitivity of our DLS/SLS analysis.

The long-term stability of the hbEVs (over 6 weeks) was checked using PBS–citrate with three different osmolarities (50, 150, and 300 mOsm/L, Figure 3h). The intensity *I* has decreased during the first week of storage but remained constant thereafter (up to 6 weeks). *R*_h_ (Figure 3h) and *ρ* (not shown) were largely conserved throughout the observation time in all of the samples. In the experiment determining the long-term preservation of hbEVs (up to 43 days) at 4 °C, the light scattering intensity decreased during the first week of storage. The analysis of the serially diluted samples showed no evidence that the hbEVs would dissolve upon dilution (Appendix D). Since the intensity of the samples shown in Figure 3h appeared to stabilize after a week of hbEVs storage, we suspect that the drop of the DLS intensity during the first week of sample storage could be due to the adhesion of some portion of the vesicles to the cuvette surface (samples were stored and measured in the same glass cuvette at all time-intervals) and not necessarily due to degradation of the hbEVs.

To test whether haemoglobin was retained in hbEVs in medium with different osmolarities (at 300, 150, or 50 mOsm/L, after 43 days of storage at 4 °C), the samples were centrifuged for 5 min at 17,500 g and 20 °C to pellet the vesicles (estimated cut off diameter size [75] was 150 nm). The centrifugation time was calculated using an online calculator [75,76] by considering the distance from the rotor axis to the sample surface level x_min_ = 90 mm, the distance from the rotor axis to the tube bottom x_max_ = 100 mm, and the water density and viscosity, which were 0.99823 g/mL and 1.0016 cP, respectively. After centrifugation, a similar amount of red pellet was visible by the naked eye in all of the samples, and no red coloration of the supernatant was observed in either of them (Appendix G, Figure A7). This supported our above indications that the hbEVs retained haemoglobin and that they were largely resistant to hypoosmolar stress down to 50 mOsm/L.

Cryo-TEM analysis of the hbEVs isolate (stored in 300 mOsm/L PBS–citrate at 4 °C) performed at 6 weeks and at more than 7 months after isolation further supports the indications that were obtained by DLS/SLS. The morphology of a large portion of hbEVs appeared to be conserved (Figure 4a–c), although some empty vesicles (Figure 4b,d) and some aggregates of the remnants of degraded vesicles were observed (Appendix B).

## 3. Discussion

EV formation is involved in the physiological and pathophysiological processes that take place in living systems. The diversity of the mechanisms that lead to the release of small particles from cells contributes to the heterogeneity of the isolated material, which often contains populations of submicron particles of variable compositions and morphologies. Nowadays, it is well known that the field of EV studies suffers from poor reproducibility and inadequacy in terms of the separation of (and appropriate assignation of specific effects to) vesicles, lipoproteins, large protein complexes/aggregates, and other assemblies of biomolecules with similar/overlapping size and physico-chemical properties [26]. The system-interfering analytics and indirectness of the evidence also limit the relevance of the studies from the perspective of pursuing natural pathways (in vivo occurring processes).

In the present study, hbEVs were used as a model to compare the detection limits of UV-Vis, FCM, and DLS to directly detect the presence of proteins (UV-Vis) or particles (FCM and DLS). The detection of the presence of contents (by whatever means) is necessary, for example, for the selection of fractions of potential interest after the separation/purification steps. In our study, we further focused on the changes in the hbEVs parameters determined by DLS/SLS, which roughly reflect the abundance, size, and cargo of hbEVs after exposure to different conditions. The preservation of vesicle integrity was considered as the conservation of the membrane surface area and the retention of the cargo inside the vesicle.

### 3.1. Mechanism of hbEV Formation and Their Morphology

In mature erythrocytes, ectovesiculation ex vivo seems to be well supported by the evidence: buds of the plasma membrane are visible on the top of the echinocyte spicules, as observed by SEM (Figure 1a), which is in agreement with previously published experiments (recently reviewed by Kralj-Iglič et al. [77]). The size and shape of these buds are consistent with the size and shape of the isolated particles observed by SEM (Figure 1c–f). The cryo-TEM images showed that these particles are surrounded by a double-lined structure that is recognized as a lipid bilayer membrane (Figure 1b), which indicates that these particles are membrane-enclosed vesicles. The results shown in this work (Figure 1), the results of previous studies of erythrocyte blebbing in the presence amphiphilic molecules [73], and the lack of evidence for any other mechanism leading to the formation of small particles in this system therefore indicate that the particles in the supernatants of the suspensions of the washed erythrocytes were formed by the outward budding of the plasma membrane. The evidence of the budding process, the vast abundance of hbEVs observed in the SEM images, their apparent similarity in size and shape, the appearance of the bilayer membrane that encloses them (as observed by cryo-TEM), and the agreement of the hbEV shapes with the calculated membrane-free energy minimum were therefore considered to be sufficient proof that these particles are sub-micron membrane-enclosed vesicles that are released into the extracellular solution through the budding of the erythrocyte plasma membrane.

The classification of hbEVs as microvesicles or as apoptotic bodies seems to be less clear since apoptosis in erythrocytes (also called eryptosis) is evidenced by the redistribution of membrane constituents between the two layers, membrane budding, and its vesiculation [78]. The description of eryptosis is focused on the interaction of the erythrocyte membrane with inflammatory molecule insults and Ca^2+^ entry following the activation of Ca^2+^-permeable cation channels [79,80]. These processes induce changes and the redistribution of the membrane constituents (in particular phosphatidylserine) and changes in the local membrane curvature that lead to programmed cell death. In contrast to senescence, which lasts nearly 120 days, eryptosis occurs within 1–48 h [79]. Ca^2+^-permeable cation channels are activated by oxidation [81], and eryptosis is thus sensitive to oxidative stress [82].

In fresh blood samples, the yield of hbEVs is low. In our case, the formation of hbEVs was triggered by highly non-physiological conditions (storage at 4 °C for 6 days, in the absence of glucose and Ca^2+^, centrifugation at several hundred g) to yield samples rich in hbEVs. Therefore, the vesiculation is not expected to match that under in vivo conditions. Since similar particles were obtained during erythrocyte storage in previous studies, regardless of induction or the inhibition of the hypothetical pathways by the sequestration of Ca^2+^ ions or restraining oxidation [70,83], and in the case of membrane perturbation by various amphiphilic compounds [73,84], the proposed mechanisms of erythrocyte ectovesiculation (Ca^2+^-dependent and independent, triggered by oxidation, or cytoskeleton dissociation) seem to share a common sequel, which influences the size and shape of the shed particles. As the sample preparation in most studies involves mixing and/or centrifugation steps, the analysis of the isolates cannot distinguish whether the fission of the vesicles was enforced by an intrinsic cascade or extrinsically by the shear stress that is imposed on the cells and their fragments during the isolation process.

On the cryo-TEM images, the hbEVs exhibit a darker interior with respect to the extravesicular solution, indicating that the particle interior was denser than the outer solution. Based on the protein profile (Appendix C), it can be concluded that the composition of the interior of the hbEVs was similar as that of the mother erythrocyte, with the main contributor to protein content being haemoglobin. Considering an average haemoglobin concentration of 330 mg/mL, the molecular weight of the haemoglobin molecule of 64,000 kg/kmol and the effective diameter of the haemoglobin molecule of 2.5 nm yields that the haemoglobin volume presents about 15% of the erythrocyte volume. Additionally, the cryo-TEM images (Figure 4b and Figure A2 and Figure A3 (Appendix B)) suggest that in the event of a membrane rupture, intravesicular material may diffuse into the surrounding solution (suggesting it is not an insoluble aggregate in and of itself). This indicates that the constitution of the hbEV content is fluid-like and does not essentially determine the erythrocyte shape or the EV shape. The shape of the erythrocyte was found to be determined by the properties of its membrane, as described by the theoretically calculated shapes of the minimal membrane-free energy that had already been determined in 1970 [49] and that has already been further elaborated upon by extensive studies ([50], reviewed in [55]). It is therefore expected that the shape of hbEVs is similarly governed by the minimisation of the membrane-free energy [77].

Figure 1c,d show that most of the vesicles that are suspended in an isosmotic solution correspond to prolate shapes. Their relative volume *v* (effective volume to area ratio *v* = (*V*^2^/36π*A*^3^)^1/2^ where *V* is the volume and *A* is the surface area of a hbEV) is smaller than the corresponding value for the sphere (= 1). Adding water to the isolate created hypoosmotic conditions in the vesicle exterior, which resulted in an expected equilibration of the chemical potential of water inside and outside the hbEVs. Consequently, it was predicted that the hbEVs expanded due to the inflow of water, resulting in an increase in their relative volume and the shape becoming more spherical (Figure 1e,f), [85]. This further supports the hypothesis that the shape of hbEVs (similar to that of erythrocytes) is determined by the minimum free energy of the membrane. In contrast, unlike in erythrocytes, the hypotonic medium did not cause the gross rupture of the hbEVs membrane or the consequent leakage of the contents (Appendix G). This may be explained by differences in the membrane composition of erythrocytes and vesicles and differences in the average membrane curvature [86] that are related to the significant difference in the size of the two structures [85].

In FCM analysis, the disproportional dependency of the detected concentration on sample dilution was observed, which was interpreted as the detection of coincidental events [87]. More vesicles were detected in the hbEVs samples that were suspended in hypotonic medium. Considering that conventional FCM (as used in this study) is believed to not to be able to detect particles that are smaller than about 400 nm, we suggest that one possible reason for the discrepancy between the FCM and the DLS/SLS results is the change in the shape of the particles that affected the FCM detection. The detection of small particles by FCM depends on properties beyond their size. The intensity of the signals is a complex function of the illumination wavelength, collection angle, particle size, morphology(structure), and chemical composition [87]. The inflation of hbEVs in a hypotonic medium can apparently cause the hbEVs to expose a dimension that can be detected by FCM, increasing the number of detected events. It might therefore be interesting to test to see if/how such a change in medium tonicity impacts the resolution of the label-free identification of extracellular vesicles, such as those proposed by van der Pol et al. [88]. It has been established that fluorescent probes and fluorescent triggering can be used to increase the sensitivity of the FCM to detect EVs [87], and powerful approaches are being developed to reveal their diversity [89]. However, fluorescent labelling itself also presents certain difficulties, and as it was not used in our study, we will not discuss it further.

### 3.2. Stability of hbEVs

Our previous analyses have shown that the concentration, shape, and size of EVs isolated from blood plasma depend on the parameters of the procedure (i.e., centripetal acceleration of the centrifuge rotor, time of centrifugation, and processing temperature) [90,91]. In order to explore the range of preparation procedures that can be used with hbEVs, we isolated hbEVs using different centripetal accelerations of the centrifuge rotor. It was observed that hbEVs were present in a large amount in pellets when applying accelerations from 4000 g for 10 min (tested for 4 mL samples in a refrigerated centrifuge, Appendix A, Figure A1a–d), while the centrifugation of a 4 mL sample for 70 min at 50,000 g seemed to be enough to pellet a vast majority of the hbEVs (Appendix A, Figure A1e–f), as very few were observed in the pellet when the supernatant from the 50,000 g step was subjected to another round of centrifugation at 100,000 g for 70 min (Appendix A, Figure A1g–h). The amorphous aggregates of the material that was found in the pellet of this last step suggest redundancy and a potentially damaging effect of applying such a strong force in the isolation of hbEVs. It was recently highlighted that a large portion of microvesicles can be lost during the low-speed centrifugation steps [92], which is in line with our observations. Due to the haemoglobin cargo, hbEVs are red, and at a high enough sample concentration, the boundary of the settling vesicles can be followed by eye. hbEVs can therefore be used to evaluate the centrifugation efficiency and to optimize the centrifugation times.

Among major changes in the integrity of the hbEVs samples, the degradation of hbEVs was noted, while the aggregation or fusion of hbEVs were not observed. Our results suggest that the morphology of EVs can be well preserved under a variety of conditions (including pH 4–10, osmolarity 50–1000 mOsm/L, and temperature up to 45 °C). We speculate that the reasons for the high stability lay in the composition of the hbEVs membrane, its high curvature [86], and small size [44,85]. It is also possible that samples were simply too diluted to observe aggregation and/or fusion [46]. The landscape of the combinations of different factors and the effects of different buffer/medium compositions remain obscure. Additionally, molecular changes and/or even small morphological changes may impact their functionality. Clearly, many questions remain to be answered about the content, transferability, and diversity of EVs [89,93,94,95].

According to SEM (Figure 1 and Figure A1 (Appendix A)), different centrifugation conditions and resuspension media can change the shape of an hbEV. Using DLS/SLS, we only detected a minor loss of the scattering intensity and no significant changes in the mean *R*_h,90_ and ρ of hbEVs in hypotonic medium (50 mOsm/L); we therefore concluded that hbEVs are fairly resistant to hypoosmolar stress. A weak decreasing trend in the mean *R*_h,90_ was observed when the osmolarity of the samples increased (Figure 2g and Figure 3a). A slight change in *R*_h,90_ could be expected due to the equilibration of the chemical potential of water inside of and outside of the hbEVs and the consequent change in the size and shape of the hbEVs. Although the intensity of scattered light depends on several sample properties, the particle size is one of the forefront factors (light scattering intensity increases with particle size according to the power law [96]). Therefore, when suitably linked to *R*_h,90_, a change in *I*_90_ can be used as a more sensitive indicator of the change in the particle size than *R*_h,90_ alone.

The DLS/SLS experiments further suggested that during a short exposure time (samples were analysed on the same day as suspended in different media), hbEVs are stable in media with osmolarities of 50–450 mOsm, in 300 mOsm medium at pH of 6–8, and at temperatures below 45 °C, as they retained a size distribution and over 85% of the scattered light intensity compared to the untreated samples. In a previous study by Fuhrmann et al. [94], hypotonic dialysis was found to cause the swelling and efficient loading of EVs from MDA-MB231 breast cancer cells with the hydrophilic porphyrins caused EV aggregation, and impaired their uptake by recipient cells. The effect of hypotonic medium on the size of hbEVs was much less pronounced in our study, but as shown in the previously mentioned paper [94], saponin was efficiently used to load EVs without altering their size, demonstrating that integrity can be disrupted without an obvious change in EV morphology. Therefore, a detailed evaluation of possible changes in the functionality of hbEVs upon changes in the conditions remains to be investigated.

The good preservation of samples stored at 4 °C is in line with the previous reports of Deville et al. [97], who also observed little change in the EV properties after storage at 4 °C, and Cheng et al. [48], who observed that the exosomes that had been stored at 4 °C had the highest concentration and showed higher levels of the representative exosome markers ALIX, HSP70, and TSG101 compared to those stored at 37, 60, −20, or −80 °C. However, the results of different published studies are very inconsistent, and the general recommendation for long-term storage still appears to be −80 °C [35,98].

### 3.3. The Power and Limitations of LS Analysis for Evaluation of EV Integrity

The DLS/SLS analyses performed on hbEVs suspended in PBS–citrate at different conditions (Figure 2 and Figure 3) indicate that the simultaneous consideration of the *I* and *R*_h_ distribution that have been normalized to the values obtained before treatment (or storage) can help to detect some subtle sample changes. Before attributing *I* to the particle concentration, possible effects of the morphological changes on scattering properties must be considered in the data interpretation. If the changes in the *R*_h_ distribution are small and cannot be clearly distinguished due to the uncertainty of the method, a suitable interrelation between *R*_h,90_ and *I*_90_ can be used to increase the sensitivity of the method. This is illustrated by the sequence of measurements of hbEVs in the media with increasing osmolarity (300–1000 mOsm/L). While the correlation functions were very similar (almost overlapping) and because the *R*_h,90_ values varied by less than 10%, the decrease in *I*_90_ was apparently gradual when the media osmolarity increased, with a final deviation of more than 30% (Figure 3a). Once the change in *I*_90_ was noted, the shifting of the correlation functions towards shorter times in higher osmolarity media was also recognized through the closer inspection of the correlation functions (see the differences in the slope and the shift of the correlation functions to longer times in Figure 3, and Figure A6 in Appendix E). The trend of the differences between samples became more evident after the correlation functions were transformed into the size distributions, although as mentioned earlier, this was not directly reflected in the change of the mean *R*_h_ of the population. Normally, such a minute difference between the measurements is considered negligible in DLS, where a deviation in *R*_h_ of up to about 10% may be expected due to the possible interference of dust particles/residual cell debris or a large variance of measurements in case of analyzing highly polydisperse or low-concentrated samples (which usually applies to EV samples). However, the alignment of our data obtained with different techniques (DLS/SLS, FCM, SEM, and cryo-TEM) and the agreement with theoretical expectations regarding the influence of topology on the light scattering properties of particles support the relevance of the observed trend and suggest that the resolution of DLS may in fact be quite good.

While we were able to analyse hbEVs with DLS/SLS at a relatively low hbEVs concentration (below 0.1 mg/mL of total protein of the sample as estimated from absorbance (Appendix D)), the sample needs to be much more concentrated to enable the detection of dissolved/released macromolecules. In our case, the multiangle assessment did not unambiguously expose topological differences between the assessed samples. However, a major loss of integrity of the hbEVs was not detected by the other methods either (e.g., by pelleting the hbEVs in different media (Appendix G) and by cryo-TEM electron microscopy (Figure 4) of the sample stored at 4 °C for more than 7 months).

To further test the power of the DLS/SLS approach in detecting loss of sample integrity, hbEVs were gradually solubilised by Triton X-100. It was expected that the presence of the surfactant at sublytic concentrations would lead to the permeabilization of the vesicle membrane [99,100], the leakage of cargo proteins, and the possible swelling of the vesicles, as has been observed previously with artificial vesicles of similar size [101]. In the present study, a gradual increase in the concentration of Triton X-100 in hbEVs suspended in PBS–citrate resulted in a gradual decrease in *I*_90_ (Figure 3d). No major structural changes in the particles were observed, the change of shape to more spherical was minute (Figure 1f,h), and the decrease in *R*_h,90_ (Figure 3d–g, Table 1) was slight. The drop in the intensity with the preservation of other scattering properties could suggest that at low concentrations, Triton X-100 was cooperatively incorporated into some of the vesicles, causing their (complete) solubilisation, while others remained more or less intact unless additional detergent was added. A possible secondary peak in the size distribution at lower *R*_h_ values (that could have pertained to smaller vesicles or released proteins from the damaged vesicles) was not observed and large aggregates were generally not detected (data not shown). It was estimated that the concentration of soluble proteins would need to be at least ~2 mg/mL to be reliably detected by DLS/SLS (Appendix F), meaning that the concentration of hbEVs would need to be much higher in order to allow the detection of the peak of the released cargo proteins.

We observed differences in the shape of hbEVs exposed to different conditions (Figure 1c–h). Therefore, the potential of the DLS/SLS approach to reveal topological differences of vesicles (shape, cargo) needs further elaboration. A laser with a relatively long wavelength (λ = 660 nm) was used in this study. In future studies, it should be determined if a light source with a shorter wavelength could improve the detection of subtle differences in the shapes of nanovesicles that were not able to be identified in this study.

## 4. Materials and Methods

### 4.1. Preparation of Vesicles

Blood was donated by the authors, two females with no record of disease. Collection was established in the morning after fasting for a minimum of 12 h overnight. A G21 needle (Microlance, Becton Dickinson, USA) and 4.5 mL evacuated tubes with trisodium citrate (BD Vacutainers, 367714A, Becton Dickinson, USA) were used. Blood was centrifuged for 10 min at 300 g and 18 °C (centrifuge Centric 400/R, Domel, Slovenia) to sediment the erythrocytes. The plasma was removed while the erythrocytes were washed three times by replacing the supernatant with PBS–citrate (137 mM NaCl, 2.68 mM KCl, 10.14 mM Na_2_HPO_4_, 1.84 mM KH_2_PO_4_, 1.03 mM Na_3_C_6_H_5_O_7_, pH 7.2). The washed erythrocytes were stored in buffer solution at 4 °C.

On day 6 after blood collection, the erythrocyte suspension was homogenized by gently inverting the tube 5–10 times. Samples were then subjected to sequential centrifugation at 500 g, 2000 g, and 4000 g, and all steps were performed for 10 min at 4 °C in the centrifuge Centric 400/R (Domel, Slovenia). After the last step, the supernatant was subjected to centrifugation at 50,000 g, for 70 min, and at 4 °C in an ultracentrifuge Beckman L8–70M with rotor SW55Ti (Thermo Fisher Scientific, USA). The pelleted vesicles were washed once via resuspension in 5 mL of PBS–citrate and through the repetition of the centrifugation process at 50,000 g, for 70 min, and at 4 °C. Pellet was resuspended in PBS–citrate to obtain hbEV concentrate, which was stored at 4 °C until analysed.

### 4.2. Preparation of Samples for Characterization and Evaluation of HbEV Stability

Chemicals were purchased from Sigma Aldrich (NaCl, KCl, Na_2_HPO_4_, KH_2_PO_4_, Triton X-100), J. T. Baker (HCl), or Carlo Erba (Na_3_C_6_H_5_O_7_ × 2 H_2_O). Chemicals were solubilized in deionized water (dH_2_O) to prepare PBS–citrate (137 mM NaCl, 2.68 mM KCl, 10.14 mM Na_2_HPO_4_, 1.84 mM KH_2_PO_4_, 1.03 mM Na_3_C_6_H_5_O_7_, pH 7.2), 4 M NaCl, 1 M NaOH, 1 M HCl, and 10 mM Triton X-100. dH2O and all of the prepared solutions were filtered through 0.2-micron filters (Chromafil RC-20/25, ref. 729030, Macherey-Nagel GmbH, Germany) and were mixed in appropriate volume ratios to obtain the desired final medium composition, as denoted in Table 1. To test effect of change of the pH, samples of hbEVs/PBS–citrate were used, and the pH was adjusted through the addition of the suitable amount of 1 M NaOH or HCl (up to a few μL per 1 mL sample). The hbEVs/PBS–citrate samples were also used to test the thermal resistance of the hbEVs and their sensitivity to solubilisation by Triton X-100.

To assess changes in the samples upon storage in different media (50, 150, or 300 mOsm/L PBS–citrate), a sample of hbEV concentrate was suspended in a suitable mixture of PBS–citrate and dH_2_O, as reported in Table 1. Samples were stored at 4 °C until analysed.

### 4.3. Scanning Electron Microscopy (SEM)

Samples of erythrocytes, hbEVs/300 mOsm/L PBS–citrate, and hbEVs/50 mOsm/L PBS–citrate were incubated for two hours in 2% OsO_4_ before they were applied on a 0.05-micron MCE filter (MF-Millipore^TM^, ref. VMWP01300). Then, the filter was taken out from the holder and was treated by changing the bath solution. After three steps of washing in distilled water, the sample was dehydrated in a graded series of ethanol (30%, 50%, 70%, 80%, 90%, absolute), treated with hexamethyldisilazane (30%, 50% mixtures with absolute ethanol, followed by pure hexamethyldisilazane), and air dried. The samples were Au/Pd coated (PECS Gatan 682) and were examined using a JSM-6500F Field Emission Scanning Electron Microscope (JEOL Ltd., Tokyo, Japan).

### 4.4. Cryo Electron Microscopy (Cryo-TEM)

Samples of hbEVs concentrate were prepared for Cryo-TEM with Vitrobot Mark IV (Thermo Fisher Scientific, Waltham, MA, US). Quantifoil^®^ R 2/2 (or 1.2/1.3), 200 (Quantifoil Micro Tools GmbH, Großlöbichau, Germany) holey carbon grids were glow discharged for 60 s at 20 mA and positive polarity in air atmosphere (GloQube^®^ Plus, Quorum, Laughton, UK). Vitrobot conditions were set to 4 °C, 95% relative humidity, blot time: 5 s, and blot force: 4. An amount of 2 µL of the sample suspension was applied to the grid, blotted, and vitrified in liquid ethane. Samples were visualized under cryo conditions with a 200 kV microscope Glacios with a Falcon 3EC detector (Thermo Fisher Scientific, Waltham, MA, US).

### 4.5. Flow Cytometry (FCM)

Samples were analysed using a flow cytometer MACSQuant Analyzer (Miltenyi Biotec, Germany) and the related software. Particles were characterized based on forward (FSC) and side (SSC) scattering signals. The following settings were employed for the measurements: FSC: 458 V with hlog scaling; SSC: 467 V with hlog scaling; and the trigger was set to SSC: 1.80. The calibration beads 3 μm and 2 μm—sized were from Miltenyi Biotec B.V. & Co. KG, Germ any (MACSQuant Calibration beads, 130-093-607) while those that were 1 μm and 0.5 μm in size were from Invitrogen, Thermo Fisher Scientific, USA (Flow cytometry Sub-micron Size Reference Kit, Green fluorescent, ref.: F13839).

### 4.6. Static (SLS) and Dynamic (DLS) Light Scattering (LS)

The dynamic (DLS) and static light scattering (SLS) measurements were performed to determine the average hydrodynamic radius *R*_h_ and the radius of gyration *R*_g_ of EVs along with the intensity of scattering light *I*, which was interpreted as a measure of EV concentration or topological change in cases of the changed particle size distribution in the hbEVs samples. Samples in different media were analysed by the Instrument 3D-DLS-SLS cross-correlation spectrometer from LS Instruments GmbH (Fribourg, Switzerland). A 100 mW DPSS laser (Cobolt Flamenco, Cobolt AB, Sweden) with a wavelength λ_0_ = 660 nm was used as a light source. Prior the measurements, samples were equilibrated in the decalin bath at 25 °C for 15 min. In single angle light scattering (saLS) experiment, the scattered light was measured at the angle *θ* = 90°. For selected samples (hbEVs/50 mOsm, hbEVs/300 mOsm, hbEVs/100μM Triton X-100, hbEVs/150μM Triton X-100) maLS—multi-angular light scattering, was measured in the angular range of 30°–150° in 10° steps. Measurements were performed for 120 s, the correlation functions and the integral time averaged intensities *I*(*θ*) ≡ *I*(*q*) (where *q* is the scattering vector, defined as *q* =(4π*n*_0_/λ_0_)sin(*θ*/2), with *n*_0_ being the refractive index of the medium, in our case estimated by the respective value for water (n_0_ = 1.33)), were collected simultaneously. The intensities of scattered light *I* were normalized with respect to the Rayleigh ratio for toluene (R, cm^−1^). The excess LS intensity of the samples (R − R_0_) was calculated as a difference of the absolute LS intensity of the samples (R) and of the solvent (R_0_). To determine *R*_h_, *R*_g,_ and the shape parameter *ρ,* mathematical analysis was conducted as described in detail in [102]. In brief, the diffusion coefficient (D) of the species in the sample was derived from the correlation function of the scattered light intensity *G*_2_(*t*), applying the Siegert’s relation [74,96] to obtain the correlation function of the scattered electric field (*g*_1_(*t*)). The inverse Laplace transformation of the correlation function was performed using the Contin algorithm [103] to yield the distribution of the relaxation times of the species in the suspension. The distributions of the relaxation times of the particles were converted into size distributions by applying the Stokes–Einstein equation (*R*_h_ = *kT*6π*η*D, where *k* is the Boltzmann constant, *T* is the absolute temperature, and *η* is the viscosity of the medium, in which the particles diffuse), assuming that the particles have a spherical shape. The concentration of the particles in the samples was assumed to be low enough to not effect particle diffusion, and the viscosity of the medium was approximated using the viscosity of water.

A multiangle assessment of LS allowed for the structure factor of the hbEVs particles *P*(*q*) = *I*(*q*)/*I*(0) to be determined, where *I*(0) is the intensity of the scattered light extrapolated to *q* (or *θ*) = 0, from which the radius of gyration, *R*_g,_ was calculated using the Guinier approximation *P*(*q*) =exp[(-*qR*_g_)^2^/3]. To estimate the topology of the particles, the results of the analysis were also expressed in the form of the so-called shape parameter *ρ* = *R*_g_/*R*_h,0_, where *R*_h,0_ is the extrapolated value of *R*_h_ to angle 0 [74].

The analysis of the thermal stability was performed by the LitesizerTM 500 instrument (Anton Paar GmbH). The samples were heated from 15 °C to 60 °C in 5 °C steps. After reaching each target temperature, the samples were equilibrated for an additional 5 min before 10 measurements that were 20 s in duration were performed. The size distributions were obtained from the mean correlation function with the corresponding program (Kalliope™, Anton Paar GmbH) using the Contin approach.

An overview of the samples that were analysed and the methods that were used is given in Table 2.

### 4.7. Estimation of Protein Content

UV-Vis spectra (180–800 nm) were measured by the Nanodrop One C (Thermo Scientific, USA). The absorbance values at the characteristic peaks of the proteins (general) at 280 nm and the haemoglobin peaks at 414 nm were used to the calculate protein content of the sample.

Sodium dodecyl sulphate polyacrylamide gel electrophoresis (SDS-PAGE) in reducing conditions was performed using a True-Page precast gel 4–20% (ref. PCG2004, Sigma Aldrich, Germany) and the suitable corresponding buffers. Erythrocyte ghosts and the soluble protein fraction were obtained by the resuspension of the erythrocytes in distilled water followed by centrifugation at 17,500 g for 10 min. Ghosts in the pellet (washed twice with distilled water) and the soluble fraction (supernatant) were collected. Before being loaded on the gel, samples of the erythrocytes, soluble fraction, ghosts, and hbEVs were mixed with a suitable amount of 4× loading buffer and 10× ditiotreitol reducer and were heated at 95 °C for 5 min.

## 5. Conclusions

Through SEM and cryo-TEM, we have observed that hbEVs t shed from the top of echinocyte spicules largely conserved their shape and size for least 6 weeks and up to 7 months after isolation from erythrocyte suspensions that had been aged in vitro. Changes in the vesicle topology in our samples as determined by the shape factor ρ were found to be minor. With the presented evaluation of the vesicle stability/integrity determined by the light scattering techniques, we suggest that DLS/SLS could be used conveniently as an integrity-check-up tool in studies of EVs and other nanovesicle-containing preparations. The DLS/SLS technique allows the analysis of hbEVs, even when they are at relatively low concentrations hbEVs, where other direct methods (FCM and UV-Vis absorbance measurements) fail to detect any samples. Using DLS/SLS, the sample is able to be fully recovered and can be subsequently used for other applications. We suggest that if using set-ups similar to ours, then the *R*_h_ distribution that is obtained using DLS at one angle should be considered together with changes in *I* and compared to the initial sample in order to assess sample preservation.

As far as we know, this is one of the first systematic assessments of the morphological changes of EVs under a variety of conditions, which is relevant to sample handling during the isolation and the application in EV studies where EV handling involves a change in the buffer, the introduction of the EVs into culture media, affinity-based separation, storage, etc., and further detailed assessment considering the impact of processing on the integrity of EVs at the molecular level is still required. Due to haemoglobin cargo, hbEVs are red, and at a high enough sample concentration, they can be followed by the naked eye, making them a particularly convenient model for establishing/validating protocols. For certain purposes, better characterization might be needed.

## Figures and Tables

**Figure 1 ijms-22-12772-f001:**
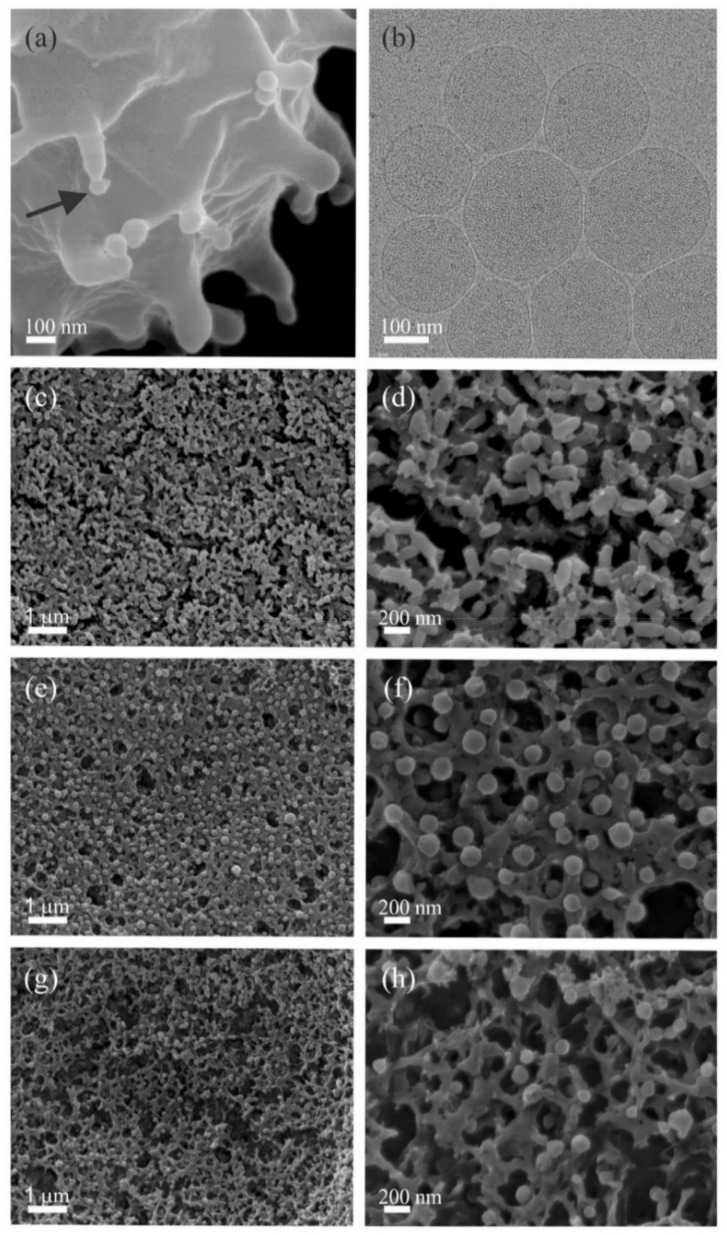
SEM (**a**,**c**–**h**) and cryo-TEM (b) images of hbEVs. (**a**) Budding echinocyte (arrow points to the bud); (**b**–**d**) isolated hbEVs in isotonic medium (PBS–citrate, 300 mOsm/L); (**e**,**f**) isolated hbEVs suspended in hypotonic medium (PBS–citrate, 50 mOsm/L); (**g**,**h**) osolated hbEVs in medium (PBS–citrate, 300 mOsm/L) with added surfactant Triton X-100 (150 μmol/L). Isolates were fixed for imaging by SEM on the filter paper. The scale bars in (**c**) and (**d**) also apply to the panels (**e**,**g**) and (**f**,**h**), respectively. Due to dehydration of the samples during preparation, the size of the EVs is expected to be larger than observed in the figures. We estimate a globular morphology with an effective radius between 100 and 200 nm. The homogeneity of the size of the hbEVs within the samples was not assessed.

**Figure 2 ijms-22-12772-f002:**
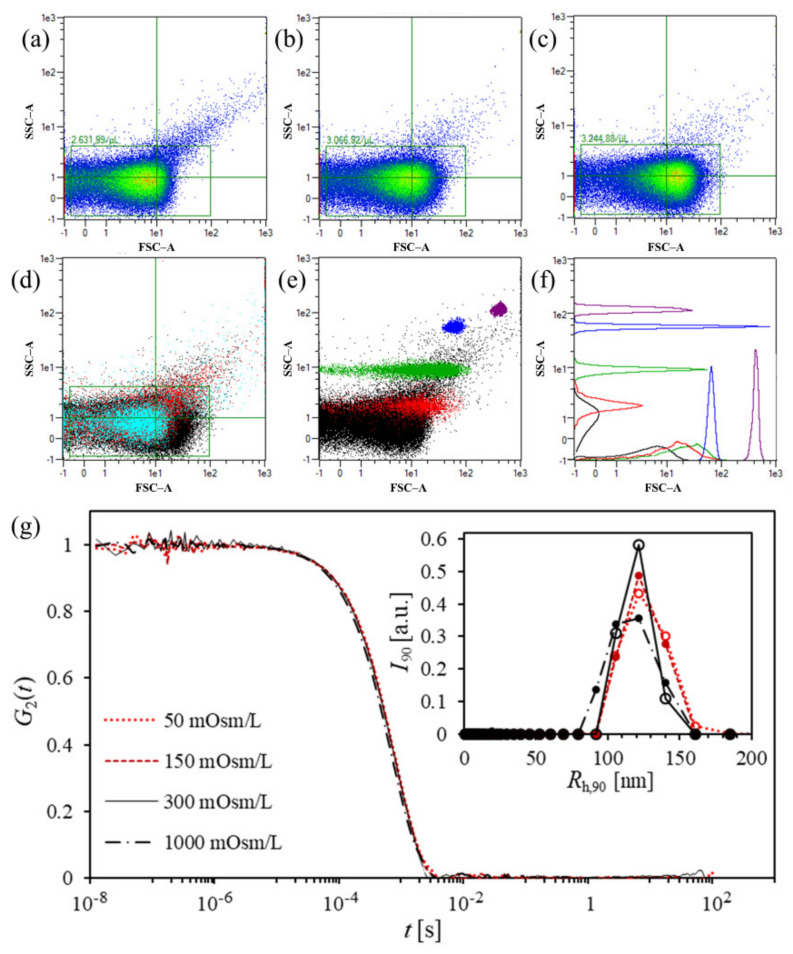
FCM and DLS/SLS analysis of hbEV isolates. (**a**–**d**) Scatterplot presenting forward (FSC–A) and side-scattering (SSC–A) signals of hbEV isolates suspended in media with different osmolarites (colour scale from blue to red pertains to increasing event density): 300 mOsm/L (a), 150 mOsm/L (**b**), 50 mOsm/L (**c**), and an overlay of scatterplots a–c (**d**): Light blue—300 mOsm/L, red—150 mOsm/L, black—50 mOsm/L. (**e**–**f**) An overlay of a scatterplot of a representative hbEV sample (300 mOsm/L PBS–citrate, black dots) and calibration beads of different sizes: red—0.5 μm, green—1 μm, blue—2 μm, and purple—3 μm; in (**f,**) distributions of events over FSC and SSC signals are presented normalized to the area under the curve; (**g**) the measured correlation functions *G*_2_(*t*) of the hbEVs in PBS–citrate with different osmolarities, as indicated in the figure. The scattered light was measured at a 90 ° angle and at a temperature of 25 °C. The corresponding distributions of the hydrodynamic radius were determined at a 90° angle, *R*_h,90_, and are presented in the inset.

**Figure 3 ijms-22-12772-f003:**
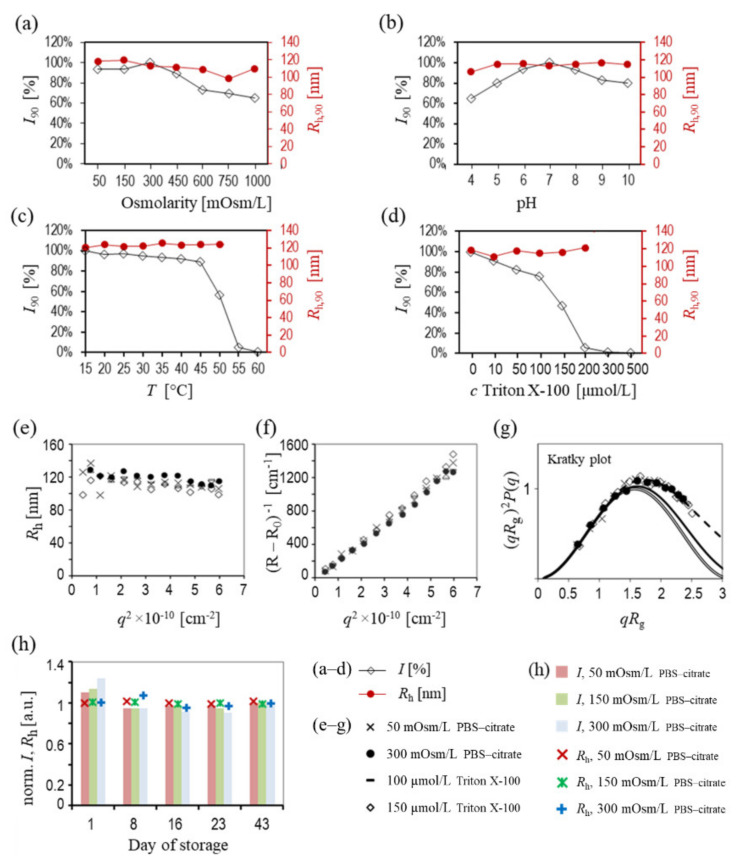
LS analysis of hbEV samples considering different osmolarities, pH values and temperatures, and different concentrations of added Triton X-100. The normalized intensity of the light scattered from the hbEVs suspension *I* measured at 90° (*I*_90_, primary axis, empty black squares) and the mean hydrodynamic radius *R*_h_ measured at 90° (*R*_h_,_90_ secondary axis, red circles) obtained from the correlation function *G*_2_(*t*) are depicted for: (**a**)—different medium osmolarities, (**b**)—different pH values of the suspension, (**c**)—different temperatures of the suspension, (**d**)—different concentrations of the added Triton X-100. In a–d, *I* was normalized to the respective value of the starting sample (a: osmolarity of the medium = 300 mOsm/L, b: pH = 7, c: T = 15 °C, d: *c*_TritonX-100_ = 0, respectively). The points are connected to facilitate the tracking of the results. Multiangle analysis of the light scattered on hbEVs at different media osmolarities and with the addition of different concentrations of Triton X-100 is shown in (**e**): angular dependence of *R*_h_, (**f**): angular dependence of the reciprocal intensity of the scattered light (R − R_0_)^−1^, and (**g**): the Kratky plot representation (dependence of (*qR*_g_)^2^*P*(*q*) on *qR*_g_) of the measured data. The curves corresponding to the theoretical structure factor ρ pertaining to the filled sphere (full line) and to the hollow sphere (double line) and the Guinier function (dashed line) are also depicted in Panel g. In (**e**–**g**)—circles: 300 mOsm/L, crosses: 50 mOsm/L, dashes: 100 μmol/L, diamonds: 150 μmol/L of added Triton X-100; h: *I*_90_ (boxes) and *R*_h,90_ (markings) pertaining to hbEVs suspended in media with different osmolarites assessed on days 1, 8, 16, 23, and 43 after isolation from an aged erythrocyte suspension that had been aged in vitro. *I*_90_ and *R*_h,90_ in Panel (**h**) are presented as normalised to the average values of the respective measurements over time. a. u.—arbitrary units. Red—50 mOsm/L, green—150 mOsm/L, blue—300 mOsm/L.

**Figure 4 ijms-22-12772-f004:**
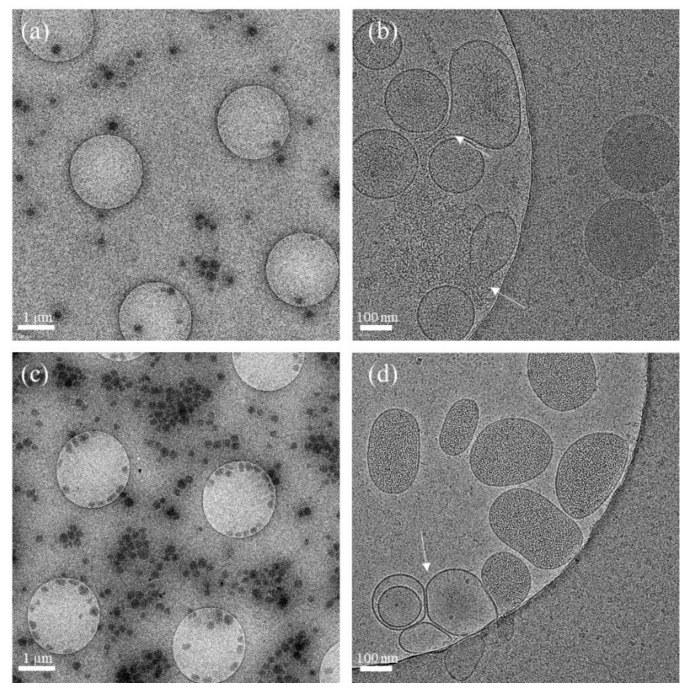
Cryo-TEM images of the hbEVs stored at 4 °C for 6 weeks (**a**,**b**) (micrographs taken from the same grid, different magnifications), and for 7 months (**c**,**d**) (micrographs taken from the same grid, different magnifications). Arrows point to some ruptured (**b**) and lighter-shaded vesicles (indicating leakage of EV cargo) (**d**).

**Table 1 ijms-22-12772-t001:** Comparison of hbEVs at different conditions as observed by DLS/SLS. Light scattering parameters: *I*_90_ normalized to the initial sample in PBS–citrate (*I*_90_/*I*_90,hbEVs/300 mOsm/L PBS–citrate_) and *ρ* pertaining to hbEVs suspended in PBS–citrate. Results are given for samples in media with different osmolarities and in presence of added Triton X-100. Samples with added Triton X-100 were prepared in 300 mOsm/L PBS–citrate.

	DLS/SLS	
Sample	*I*_90_/*I*_90_,_hbEVs/300 mOsm/L PBS–citrate_	*ρ*
hbEVs/300 mOsm/L PBS–citrate	100%	0.75
hbEVs/50 mOsm/L PBS–citrate	94%	0.76
hbEVs/100 μmol/L Triton X-100	76%	0.77
hbEVs/150 μmol/L Triton X-100	47%	0.81

**Table 2 ijms-22-12772-t002:** Sample preparation.

Sample Designation	PBS–citrate (μL)	dH2O (μL)	4 M NaCl (μL)	HbEVs Isolate (μL)	Methods	Assessment Focus
hbEVs/50 mOsm/L (PBS–citrate)	-	833.0		167.0	maLS, SEM	Effect of medium osmolarity, storage
hbEVs/150 mOsm/L	-	820.5	12.5	167.0	saLS	Effect of medium osmolarity
hbEVs/150 mOsm/L PBS–citrate	333.0	500.0	-	167.0	maLS	storage
hbEVs/300 mOsm/L	-	802.0	31.0	167.0	saLS	Effect of medium osmolarity
hbEVs/300 mOsm/L PBS–citrate	833.0	-	-	167.0	maLS, SEM	Thermal and pH resistance, storage, solubilisation by Triton X-100
hbEVs/450 mOsm/L	-	783.0	50.0	167.0	saLS	Effect of medium osmolarity
hbEVs/600 mOsm/L	-	764.5	68.5	167.0	saLS	Effect of medium osmolarity
hbEVs/750 mOsm/L	-	745.5	87.5	167.0	saLS	Effect of medium osmolarity
hbEVs/1000 mOsm/L	-	718.5	114.5	167.0	saLS	Effect of medium osmolarity

Osmolarity of the samples was estimated from molarity of used solutions. maLS—multi-angular light scattering; saLS—single angle light scattering (90°angle).

## Data Availability

Data is contained within the article and/or appendixes.

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
