# Peer review of "Stability of Erythrocyte-Derived Nanovesicles Assessed by Light Scattering and Electron Microscopy"

_ijms, 2021, doi:10.3390/ijms222312772_

Round 1

Reviewer 1 Report

In this manuscript, the authors investigated the stability of hbEVs vesicles. The following comments should be addressed:

- The introduction lacks recent examples of existing work on this same topic. This limits the perception of the impact and importance of the manuscript.

- The manuscript abstract and introduction need extensive re-editing and revision of the English language and style. The text is not fluent, words are missing, and the language and style do not reflect usual English standards.

- The acronyms are not all explained in the text.

- The authors declare that on cryo-EM, the EVs shows a dark content indicating a rich filling (e.g. Fig. 1b and Fig. 4). However, this dark content is not visible on low magnification. For example, in Fig 4a and c the EVs have higher contrast than the grid and ice background. However, in Fig 4b and d, the contrast is higher for the grid background. Are the images showing the same sample?

- scale bar should be shown in all images.

- EVs are too small in size to be detected by flow cytometry. The EVs size detected via EM does not match with the FCM size profile.

- Fig 2g. Is Rh standing for the hydrodynamic radius? The correlation function does not show any change in size at different osmolarities while the authors claim otherwise. This should be clarified. Also, the correlation function for all samples should be shown.

- Fig 3 is a low-resolution image and it is not possible to read the axis and legends.

- Fig. 3 misses most of the legends.

- Fig 3. Some of the intensity profiles go above 100%. How do the authors explain this?

- The authors should provide (or speculate over) the physical-chemical mechanism underlying the stability under the conditions (pH, osmolarity, detergent) investigated.

- the manuscript is lacking previous literature about the investigation of EVs stability.

Author Response

We thank the Reviewer for very constructive critics and her/his suggestions for improving of our article. In an extensive revision of the text the following changes were made:

  • Abstract and Introduction were completely rewritten
  • Figures 1, 3, 4 and A4 were adapted and/or improved in resolution.
  • Another Figure was prepared (Figure A6) and a chapter was added into the Appendix section (Appendix E) to show correlation functions requested by the Reviewer 1.
  • Discussion was upgraded with suggestions of reviewers
  • 54 references were added.
  • Minor language corrections have been made throughout the text

The changes in the manuscript are marked with the “track changes”. Our responses to the reviewer's comments are given below. The given line numbers refer to the “simple markup” view of the reviewed text.

Reviewer 1: “ In this manuscript, the authors investigated the stability of hbEVs vesicles. The following comments should be addressed

- The introduction lacks recent examples of existing work on this same topic. This limits the perception of the impact and importance of the manuscript.”

Response:

The introduction was rewritten and the background and literature review was extended. The references [19-29], [33-48] are added to illustrate the relevance of the topic.

Reviewer 1: “- The manuscript abstract and introduction need extensive re-editing and revision of the English language and style. The text is not fluent, words are missing, and the language and style do not reflect usual English standards.”

Response:

The introduction was rewritten and revised by a native English speaking scientist dr. Mitja Drab.

Reviewer 1: “- The acronyms are not all explained in the text.”

Response:

In the first version of the manuscript, acronyms explained in abstract were not explained again in the main text. We added additional explanations to their first appearance in the main text.

Reviewer 1:

“- The authors declare that on cryo-EM, the EVs shows a dark content indicating a rich filling (e.g. Fig. 1b and Fig. 4). However, this dark content is not visible on low magnification. For example, in Fig 4a and c the EVs have higher contrast than the grid and ice background. However, in Fig 4b and d, the contrast is higher for the grid background. Are the images showing the same sample?”

Response:

The images show aliquots of the same sample, but at different time points of storage (Fig4a,b after 6 weeks, Fig c,d: after 7 months), the pair a,b was collected on a single EM grid and the pair c,d was taken on another (single) grid. This clarification was added to the Figure caption in the manuscript, lines 323-324). There are multiple possible reasons for the difference in contrast. Contrast can be different because the images were acquired at different defocus values and also the ice thickness can vary (between the samples and across the grid). However, we think that in this case, the main cause was due to the fact that images presented in the panels a) and b), of the original Figure 4, have been recorded with automatic histogram adjustment. For those in panels c) and d) the automatic histogram adjustment has been disabled. This discrepancy was unintentional and unfortunately this detail was missed during manuscript preparation. We therefore thank the reviewer for his/her caution. We adapted the Figure 4 to show comparably histogram adjusted images (All the required images can be provided to the reviewer if requested). To our opinion, presence of contents of the EVs, can now be recognized in both high-magnification images. The dark is replaced by “darker” and the adjective "rich" is omitted (lines 173-174).

Reviewer 1: “- scale bar should be shown in all images.”

Response:

Scale bars are added to the Figure 1 (Panels e-h).

Reviewer 1: “- EVs are too small in size to be detected by flow cytometry. The EVs size detected via EM does not match with the FCM size profile.”

Response:

We thank the reviewer for exposing her/his observation that “EVs size detected via EM does not match with the FCM size profile,” and we would kindly ask for a further explanation of recognizing this mismatch. We completely agree (as stated in the manuscript, line 423)) that the size of hbEVs as revealed by cryo-EM and SEM (diameter mostly below 300 nm) seems to be too small for detection by conventional FCM. However, please note that in SEM preparation, samples are dehydrated which results in some shrinking. The FCM data presented in the paper is the one directly obtained on the described samples (using settings described in detail in the Method section). Detection of small particles by FCM depends on their scattering properties (beyond their size), and it is well established that the size can only be very roughly estimated from the FSC/SSC signals. We would also like to stress that the shape of the event clouds depends on the settings of the lasers and scaling in the plots (for that reason, results of the analysis of the standard beads in the same settings are shown for comparison in Figure 2). To our knowledge and experience, “matching” of the scatterplot signals with EVs size is quite elusive as the intensity of the signals is a complex function of particle size, morphology(structure) and chemical composition [87], which is further pronounced in the cases when the wavelength of the light source approaches the size of the analysed particle (as in case of EVs and the usual set of FCM lasers). We provided an interpretation that seemed plausible to us in the lines 198-209 and 419-430.

[87] Nolan, J.P. Flow Cytometry of Extracellular Vesicles: Potential, Pitfalls, and Prospects. Curr Protoc Cytom 2015, 73, 13.14.11-13.14.16, doi:10.1002/0471142956.cy1314s73.

Reviewer 1:

“- Fig 2g. Is Rh standing for the hydrodynamic radius? The correlation function does not show any change in size at different osmolarities while the authors claim otherwise. This should be clarified. Also, the correlation function for all samples should be shown.”

Response:

True, Rh stands for hydrodynamic radius, the explanation of the abbreviation is added to the line 196-197. As noted by the reviewer, the changes in the correlation functions are hardly observed, and were at first overlooked also by us. We therefore did not see any advantage of showing all the measured correlation functions (Please note that more than 1000 correlation functions were measured and manually analysed), as that would rather dim the detected trend of the shift (they all look “the same”). We therefore decided to keep the selected 4 examples in the main text to support the evidenced trend (the extreme 2, the physiological one (300 mOsm/L) and one more (150 mOsm/L), which was also assessed for a long-term stability). The trend in such presentation can be visible to an attentive reader from the order of the data lines, established in the slope and developed in the lag of the four presented correlation functions. The trend of the difference between the samples is further exposed after the conversion of correlation functions into the size distribution, while it may not necessarily be obviously reflected in the change of mean Rh of the population. Normally such a minute difference between the measurements is considered negligible in DLS, where about 10% variance may be expected due to the interference of possible dust particles/residual of cell remnants or large variance of measurements in case of highly polydispersed or low concentrated samples (usually applying to the EV samples). However, we believe that alignment of our data obtained by different techniques (DLS/SLS, FCM and EM), and the agreement with the theoretical expectations of the effect of topology on the light scattering properties of the particles support the relevance of the detected trend. We further added the correlation functions obtained for samples analysed in the range of osmolarities 50-1000 mOsm/L, pH 4-10, temperatures 15-60°C, Triton X-100 treatment, and 43-day storage at 4°C (Figure A6, Appendix E). Please note, that the correlation functions of four samples presented in the main text and the sequence of the samples 50-1000 mOsm/L presented in Appendix E resulted from the separate experiments with hbEVs of different donors. We hope that correlation functions added to the SM section satisfy the reviewer’s claim.

Reviewer 1: “- Fig 3 is a low-resolution image and it is not possible to read the axis and legends.”

Response:

The Figure 3 is adapted for better resolution and the legend is added.

Reviewer 1: “- Fig. 3 misses most of the legends.”

Response:

The Legend is added to Figure 3.

Reviewer 1: “- Fig 3. Some of the intensity profiles go above 100%. How do the authors explain this?”

Response:

We did not observe any profile exceeding 100 % intensity in the tests assessing hbEVs in different medium (Figure 3a-d, black squares). We speculate that the reviewer had in mind the normalized data of monitoring samples in different media for 43 days (Figure 3h). The intensity of light scattering was indeed higher at first assessment. Some minor fluctuations of the measured intensity data were observed in the further measurements as well. As analysis of serially diluted samples did not expose any evidence for hbEVs dissolving upon dilution, as the intensity of the sample presented in Fig3h apparently stabilized after a week of hbEVs storage, and as the EM analysis proved presence of the hbEVs after more than 6 months, we suspect that the drop of the scattering intensity in the first week of sample storage could be due to few larger aggregates or cell remnants in the samples (not affecting the size distribution as aggregates were not detected as visible from the correlation functions added to the supplementary material (Figure A6, Appendix E), but possibly contributing to the higher measured intensity). The other plausible explanation could be that of some of the vesicles adhered to the cuvette surface (samples were stored and measured in the same glass cuvette at all time-points). We therefore decided to normalize the data to the average of all 5 time-point measurements and not to the initial value, resulting in the initial value of the intensity over 1. The explanation is added to the discussion lines 295-306.

Reviewer 1: “- The authors should provide (or speculate over) the physical-chemical mechanism underlying the stability under the conditions (pH, osmolarity, detergent) investigated.”

Response:

Our speculations are added into the discussion lines 459-464, 543-546.

Reviewer 1: “- the manuscript is lacking previous literature about the investigation of EVs stability.”

Response:

The reviewers note is acknowledged and additional paragraphs were added (lines – 79-108) and the reference list is supplemented with the following references are added:

[35] Yuan, F.; Li, Y.M.; Wang, Z. Preserving extracellular vesicles for biomedical applications: consideration of storage stability before and after isolation. Drug Deliv 2021, 28, 1501-1509, doi:10.1080/10717544.2021.1951896.

[48] Cheng, Y.; Zeng, Q.; Han, Q.; Xia, W. Effect of pH, temperature and freezing-thawing on quantity changes and cellular uptake of exosomes. Protein & Cell 2019, 10, 295-299, doi:10.1007/s13238-018-0529-4.

[97] Deville, S.; Berckmans, P.; Van Hoof, R.; Lambrichts, I.; Salvati, A.; Nelissen, I. Comparison of extracellular vesicle isolation and storage methods using high-sensitivity flow cytometry. PLOS ONE 2021, 16, e0245835, doi:10.1371/journal.pone.0245835.

[98] Jeyaram, A.; Jay, S.M. Preservation and Storage Stability of Extracellular Vesicles for Therapeutic Applications. Aaps j 2017, 20, 1, doi:10.1208/s12248-017-0160-y.

[36] Jacobsohn, M.K.; Bazilian, L.S.; Hardiman, J.; Jacobsohn, G.M. Effect of pH on the affinity of phospholipids for cholesterol. Lipids 1989, 24, 375-382, doi:10.1007/bf02535144.

[37] Tegmo-Larsson, I.M.; Hofmann, K.P.; Kreutz, W.; Yatvin, M.B. The effect of pH on vesicles composed of phosphatidylcholines and N-acylamino acids: A calcein release fluorescence study. Journal of Controlled Release 1985, 1, 191-196, doi:https://doi.org/10.1016/0168-3659(85)90017-3.

[38] Leung, C.-Y.; Palmer, L.C.; Kewalramani, S.; Qiao, B.; Stupp, S.I.; Olvera de la Cruz, M.; Bedzyk, M.J. Crystalline polymorphism induced by charge regulation in ionic membranes. Proceedings of the National Academy of Sciences 2013, 201316150, doi:10.1073/pnas.1316150110.

[39] Schlieper, P.; Steiner, R. Effect of pH and different substrates on the electrokinetic properties of (Na+, K+)-ATPase vesicles. Biophys Struct Mech 1983, 9, 193-206, doi:10.1007/bf00537816.

[40] Zheng, L.-Q.; Shui, L.-l.; Shen, Q.; Li, G.-Z.; Baba, T.; Minamikawa, H.; Hato, M. pH and salt-induced reversible aggregation of nonionic synthetic glycolipid vesicles. Colloids and Surfaces A: Physicochemical and Engineering Aspects 2002, 207, 215-221, doi:https://doi.org/10.1016/S0927-7757(02)00118-8.

[41] Mondal Roy, S.; Sarkar, M. Membrane fusion induced by small molecules and ions. Journal of lipids 2011, 2011, 528784-528784, doi:10.1155/2011/528784.

f42] Thureson-Klein, A.; Klein, R.L.; Chen Yen, S.H. Morphological effects of osmolarity on purified noradrenergic vesicles. Journal of neurocytology 1975, 4, 609-627, doi:10.1007/bf01351540.

[43] De Michelis, M.I.; Pugliarello, M.C.; Rasi-Caldogno, F.; De Vecchi, L. Osmotic Behaviour and Permeability Properties of Vesicles in Microsomal Preparations from Pea Internodes. Journal of Experimental Botany 1981, 32, 293-302.

[44] Ohki, S. Effects of divalent cations, temperature, osmotic pressure gradient, and vesicle curvature on phosphatidylserine vesicle fusion. J Membr Biol 1984, 77, 265-275, doi:10.1007/bf01870574.

[45] Ibarguren, M.; Bomans, P.H.; Ruiz-Mirazo, K.; Frederik, P.M.; Alonso, A.; Goñi, F.M. Thermally-induced aggregation and fusion of protein-free lipid vesicles. Colloids Surf B Biointerfaces 2015, 136, 545-552, doi:10.1016/j.colsurfb.2015.09.047.

[46] Eum, K.M.; Riedy, G.; Langley, K.H.; Roberts, M.F. Temperature-induced fusion of small unilamellar vesicles formed from saturated long-chain lecithins and diheptanoylphosphatidylcholine. Biochemistry 1989, 28, 8206-8213, doi:10.1021/bi00446a036.

[47] Howard, F.B.; Levin, I.W. Lipid vesicle aggregation induced by cooling. Int J Mol Sci 2010, 11, 754-761, doi:10.3390/ijms11020754.

[48] Cheng, Y.; Zeng, Q.; Han, Q.; Xia, W. Effect of pH, temperature and freezing-thawing on quantity changes and cellular uptake of exosomes. Protein & Cell 2019, 10, 295-299, doi:10.1007/s13238-018-0529-4.

Reviewer 2 Report

The authors submitted a comparative study on the size of particles deriving from erythrocyte suspensions in vitro over a period of a few days. They then isolated these particles by differential centrifugation and studied their morphology and the effect of reagents on, using separate protocol over a period of 7 months.

The work provides some hints about the applicability of initial EVs characterization protocols, as suggested by MISEV. However, due to the need for protocol harmonization, it seems difficult to understand how this work reflects other groups data, and how several labelling protocols could take benefit.

Many groups (had to) and tried to characterize EVs content. There is some general information, such as very low protein content, and some evidence about Hg presence. According to the guidelines, and because of the need of harmonization, it should be necessary to have relevant information.

Therefore, it is further difficult to understand how this work could prove erythrocyte EVs as a reference material.

So far, it is unclear how this work could be improved in order to show parallels between cell aging and EVs shedding, or any other physiological process. It is also unclear how the methodological approaches may be informative to those working in the field.

There are multiple minor language errors, “the” is usually missing as well. Please find below some specific comments:

L20 Please revise the first sentence to fit abstract format as a continuous text.

L30 remained consistent for at least a period of X months at 4oC?

L32 I was decreased at least by 20%.

L33 due to the simplicity of preparation protocols, and the homo… in a wide …?

L43 revise potentials

L45 The sentence is quite confusing and similar to the previous half. Please revise or better specify.

L46 the reproducibility

L47 Please provide reference and revise

L48 burdening

L49 analytics?

L53 The International…

L55 remove to the sample

L58 What is the meaning of “to the satisfactory level”?

L65 What is the meaning of “internal framework”?

Regarding erythrocytes morphology there are many explanations, from Hg functionality to shear stress. On Donnan effect and relevant misconceptions there is an extensive educational text by Alex Yartsev:

https://derangedphysiology.com/main/cicm-primary-exam/required-reading/cellular-physiology/Chapter%20121/gibbs-donnan-effect

Better consider this ref:

Stomatocyte–discocyte–echinocyte sequence of the human red blood cell: Evidence for the bilayer– couple hypothesis from membrane mechanics. Lim H. W et al, 2002; DOI: 10.1073/pnas.202617299

and describe the cause and physiological role of these transformations in vivo.

I would suggest the introduction to be oriented around details. Going from the vague reference to clinical applications to stored blood (aimed for transfusion) and finally to the aged cells is quite problematic. Please follow a trend and cover the relevant fields (physiology, storage conditions, time etc). If no information is available, please state so.

There are two recent papers regarding transformation in association with storage and it is pretty clear that there are concerns about cell viability-functionality, e.g.

https://www.mdpi.com/2076-3417/10/9/3209

https://www.karger.com/Article/FullText/508711

There are several technical questions about EVs clustering of diode discriminative ability. Please advise

Nolan, J.P. 2015. Flow cytometry of extracellular vesicles: potential,pitfalls, and prospects.Curr. Protoc. Cytom.73:13.14.1-13.14.16.doi: 10.1002/0471142956.cy1314s73 (this ref has been used in the Discussion, but the authors did not comment step by step, comparing Nolan suggestions and experimental approaches)

Edwin van der Pol et al 2018. Absolute sizing and label-free identification of extracellular vesicles by flow cytometry, Nanomedicine: Nanotechnology, Biology and Medicine. https://doi.org/10.1016/j.nano.2017.12.012

Did the authors used any “staining”during cryomicroscopy?

L344 I also don’t understand the concept of leakage. Obviously there are many questions about EVs content and transfer, but in the case of erythrocytes, it is possible that diffusion may cause the formation of these more stable structures [Marcoux, G., Duchez, AC., Cloutier, N. et al. Revealing the diversity of extracellular vesicles using high-dimensional flow cytometry analyses. Sci Rep 6, 35928 (2016). https://doi.org/10.1038/srep35928; 20G. Fuhrmann, A. Serio, M. Mazo, R. Nair, M. M. Stevens, J. Control. Release 2015, 205, 35;  25M. Magnani, L. Rossi, M. D'ascenzo, I. Panzani, L. Bigi, A. Zanella, Biotechnol. Appl. Biochem. 1998, 28, 1]

Author Response

We are grateful to the Reviewer for a very constructive critic and her/his suggestions for improving of our article. In an extensive revision of the text the following changes were made:

  • Abstract and Introduction were completely rewritten
  • Figures 1, 3, 4 and A4 were adapted and/or improved in resolution.
  • Another Figure was prepared (Figure A6) and a chapter was added into the Appendix section (Appendix E) to show correlation functions requested by the Reviewer 1.
  • Discussion was upgraded with suggestions of reviewers
  • 54 references were added.
  • Minor language corrections have been made throughout the text

The changes in the manuscript are marked with the “track changes”. Our responses to the reviewer's comments are given below. The given line numbers refer to the “simple markup” view of the reviewed text.

Reviewer 2: “The authors submitted a comparative study on the size of particles deriving from erythrocyte suspensions in vitro over a period of a few days. They then isolated these particles by differential centrifugation and studied their morphology and the effect of reagents on, using separate protocol over a period of 7 months.

The work provides some hints about the applicability of initial EVs characterization protocols, as suggested by MISEV. However, due to the need for protocol harmonization, it seems difficult to understand how this work reflects other groups data, and how several labelling protocols could take benefit.”

Response:

We thank the reviewer for revealing the lack of explanation on importance of our study. This is addressed in the new introduction (lines – 60-108). Several studies displayed the lack of reproducibility of EV experiments. In one of the most recent reports, Torres et al. 2021 [22]  on one hand suggest: “We advise strict adherence to the published protocols by Théry et al. 42”…) but in continuation argue: “our proof-of-concept inter-laboratory study indicates significant equipment- and operator-dependent technical variability in ultracentrifugation-based EV isolation. Reproducing isolation protocols based on centrifugation speed and run time alone is clearly insufficient, and even if the k-factor is appropriately considered, variability is reduced but not eliminated.” And further “These findings prompt several questions including how reproducibility is best quantified, which criteria should be assessed, and how sensible thresholds for inter-user variability might be established.”.  We believe that studies like ours support the development of the reproducible (and “harmonized”) approaches in the very core of the problem – the lack of understanding of the complex transformation of the biological material after sample collection – by deepening understanding of (cell and) vesicle behaviour upon various conditions, and therefore expose which are the critical steps (or parameters) in the sample handling that impair reproducibility of the established protocols.

In the present work we did not deal with labelling of vesicles. We considered imaging good enough proof for the presence of vesicles (and the absence of other kinds of particles of similar size) in our samples. Our data on the semi-long period preservation of EV structure at 4°C is in line with several previous works (discussion added in lines 492- 496). We agree that study could be upgraded with some labelling, however, this was out of scope of the study.

The following relevant references are included:

[19] Nieuwland, R.; Falcón-Pérez, J.M.; Théry, C.; Witwer, K.W. Rigor and standardization of extracellular vesicle research: Paving the road towards robustness. In J Extracell Vesicles; 2020; Volume 10, p. e12037.

[20] Meng, W.; He, C.; Hao, Y.; Wang, L.; Li, L.; Zhu, G. Prospects and challenges of extracellular vesicle-based drug delivery system: considering cell source. Drug Deliv 2020, 27, 585-598, doi:10.1080/10717544.2020.1748758.

[21] Kennedy, T.L.; Russell, A.J.; Riley, P. Experimental limitations of extracellular vesicle-based therapies for the treatment of myocardial infarction. Trends Cardiovasc Med 2021, 31, 405-415, doi:10.1016/j.tcm.2020.08.003.

[22] Torres Crigna, A.; Fricke, F.; Nitschke, K.; Worst, T.; Erb, U.; Karremann, M.; Buschmann, D.; Elvers-Hornung, S.; Tucher, C.; Schiller, M.; et al. Inter-Laboratory Comparison of Extracellular Vesicle Isolation Based on Ultracentrifugation. Transfus Med Hemother 2021, 48, 48-59, doi:10.1159/000508712.

[23] Maas, S.L.; de Vrij, J.; van der Vlist, E.J.; Geragousian, B.; van Bloois, L.; Mastrobattista, E.; Schiffelers, R.M.; Wauben, M.H.; Broekman, M.L.; Nolte-'t Hoen, E.N. Possibilities and limitations of current technologies for quantification of biological extracellular vesicles and synthetic mimics. J Control Release 2015, 200, 87-96, doi:10.1016/j.jconrel.2014.12.041.

[24] Vogel, R.; Savage, J.; Muzard, J.; Camera, G.D.; Vella, G.; Law, A.; Marchioni, M.; Mehn, D.; Geiss, O.; Peacock, B.; et al. Measuring particle concentration of multimodal synthetic reference materials and extracellular vesicles with orthogonal techniques: Who is up to the challenge? J Extracell Vesicles 2021, 10, e12052, doi:10.1002/jev2.12052.

[25] Taylor, D.D.; Shah, S. Methods of isolating extracellular vesicles impact down-stream analyses of their cargoes. Methods 2015, 87, 3-10, doi:10.1016/j.ymeth.2015.02.019.

[26] Allelein, S.; Medina-Perez, P.; Lopes, A.L.H.; Rau, S.; Hause, G.; Kölsch, A.; Kuhlmeier, D. Potential and challenges of specifically isolating extracellular vesicles from heterogeneous populations. Sci Rep 2021, 11, 11585, doi:10.1038/s41598-021-91129-y.

[27] Brennan, K.; Martin, K.; FitzGerald, S.P.; O’Sullivan, J.; Wu, Y.; Blanco, A.; Richardson, C.; Mc Gee, M.M. A comparison of methods for the isolation and separation of extracellular vesicles from protein and lipid particles in human serum. Scientific Reports 2020, 10, 1039, doi:10.1038/s41598-020-57497-7.

[28] Veerman, R.E.; Teeuwen, L.; Czarnewski, P.; Güclüler Akpinar, G.; Sandberg, A.; Cao, X.; Pernemalm, M.; Orre, L.M.; Gabrielsson, S.; Eldh, M. Molecular evaluation of five different isolation methods for extracellular vesicles reveals different clinical applicability and subcellular origin. J Extracell Vesicles 2021, 10, e12128, doi:10.1002/jev2.12128.

[29] Tiruvayipati, S.; Wolfgeher, D.; Yue, M.; Duan, F.; Andrade, J.; Jiang, H.; Schuger, L. Variability in protein cargo detection in technical and biological replicates of exosome-enriched extracellular vesicles. PLoS One 2020, 15, e0228871, doi:10.1371/journal.pone.0228871.

 [35] Yuan, F.; Li, Y.M.; Wang, Z. Preserving extracellular vesicles for biomedical applications: consideration of storage stability before and after isolation. Drug Deliv 2021, 28, 1501-1509, doi:10.1080/10717544.2021.1951896.

[48] Cheng, Y.; Zeng, Q.; Han, Q.; Xia, W. Effect of pH, temperature and freezing-thawing on quantity changes and cellular uptake of exosomes. Protein & Cell 2019, 10, 295-299, doi:10.1007/s13238-018-0529-4.

[91] Larson, M.C.; Hogg, N.; Hillery, C.A. Centrifugation Removes a Population of Large Vesicles, or "Macroparticles," Intermediate in Size to RBCs and Microvesicles. Int J Mol Sci 2021, 22, doi:10.3390/ijms22031243.

[96] Deville, S.; Berckmans, P.; Van Hoof, R.; Lambrichts, I.; Salvati, A.; Nelissen, I. Comparison of extracellular vesicle isolation and storage methods using high-sensitivity flow cytometry. PLOS ONE 2021, 16, e0245835, doi:10.1371/journal.pone.0245835.

[97] Jeyaram, A.; Jay, S.M. Preservation and Storage Stability of Extracellular Vesicles for Therapeutic Applications. Aaps j 2017, 20, 1, doi:10.1208/s12248-017-0160-y.

Reviewer 2: “Many groups (had to) and tried to characterize EVs content. There is some general information, such as very low protein content, and some evidence about Hg presence. According to the guidelines, and because of the need of harmonization, it should be necessary to have relevant information. Therefore, it is further difficult to understand how this work could prove erythrocyte EVs as a reference material.”

Response:

The aim of our study was to elaborate the dynamics (sensitivity) of a model natural vesicular assembly (as a structure) under change of (some of the most elementary) external conditions that can change the equilibrium of a biological sample. Generally, in EV studies, EVs are perceived and manipulated as some rigid solids, and little attention is being paid to potential change of their identity. Yet, transiency of their functionality show otherwise. As far as we know, this is one of the first systematic assessment of EV change upon a wide set of conditions, being relevant for sample handling upon isolation and application in any EV study (change of media/buffer solution, introduction into a culture media, affinity based separation, storage etc.). Although the aim of our study was not proving of hbEVs suitability as a reference material, but rather to assess power of DLS/SLS approach to detect changes in vesicles, the evidence on their stability (which was to us surprising), easy preparation and fair monodispersity (which is quite unique among EV isolates) seemed to us a good ground to consider them to be a relevant candidate for such a purpose. In the present study hbEVs were used as a model/reference to define limits of detection of a few direct methods (that we had on hand in our lab) for assessing presence of contents of the samples (UV-vis protein detection, FCM particle detection and DLS particle detection) in our lab. It is very handy to have a quick method on hand when optimizing any procedures, e.g. for assessing which fractions contain purified vesicles. E.g. we now know, that our DLS method can detect presence of nanoparticles at much lower concentration than can be detected by direct protein detection through UV-vis or conventional flow cytometry. To continue with examples, hbEVs can be used to compare sedimentation efficiency in various rotors and centrifuges, or to optimize time of centrifugation (at high enough sample concentration, the boundary of settling EVs can be followed by eye), they also can be used to set up/validate protocols for enzymatic digestion, or suit as a model for proof-of concept e.g. in development of EV functionalization approaches etc…Each reference material has its limits of application and the same is the case with hbEVs, However, we have found them useful in many applications in our lab (we probably do not need to mention they are much cheaper than commercially available alternatives), and we wanted to share the data we collected on them. The characterization of starting material always reflects the focus of the study, we therefore agree, that hbEVs could be characterized in a much further detail, including exact composition and labelling. However, the further characterization/evaluation of the suitability is up to any possible interested parties/users.

Reviewer 2: “There are multiple minor language errors, “the” is usually missing as well. Please find below some specific comments:”

Response: The introduction was rewritten with consideration of reviewer’s comments.

“L20 Please revise the first sentence to fit abstract format as a continuous text.”

Response: The sentence was changed.

“L30 remained consistent for at least a period of X months at 4oC?”

Response: The suggestion of reviewer is accepted and conserved is replaced with consistent in the line 30.

“L32 I was decreased at least by 20%.”

Response: The suggestion of reviewer is accepted

“L33 due to the simplicity of preparation protocols, and the homo… in a wide …?”

Response: The suggestion of reviewer is accepted

“L43 revise potentials”

Response: Revised

“L45 The sentence is quite confusing and similar to the previous half. Please revise or better specify.”

Response: The sentence is removed

“L46 the reproducibility”

Response: Spelling corrected

“L47 Please provide reference and revise”

Response: Revised

“L48 burdening”

Response: The sentence was removed

“L49 analytics?”

Response: Analytics is replaced by analysis

“L53 The International…”

Response: Suggestion implemented

“L55 remove to the sample”

Response:  “to the sample” was deleted.

“L58 What is the meaning of “to the satisfactory level”?”

Response: The sentence was rephrased.

“L65 What is the meaning of “internal framework”?”

Response: Framework is replaced by scaffold.

Reviewer 2: “Regarding erythrocytes morphology there are many explanations, from Hg functionality to shear stress. On Donnan effect and relevant misconceptions there is an extensive educational text by Alex Yartsev:

https://derangedphysiology.com/main/cicm-primary-exam/required-reading/cellular-physiology/Chapter%20121/gibbs-donnan-effect

Better consider this ref:

Stomatocyte–discocyte–echinocyte sequence of the human red blood cell: Evidence for the bilayer– couple hypothesis from membrane mechanics. Lim H. W et al, 2002; DOI: 10.1073/pnas.202617299

and describe the cause and physiological role of these transformations in vivo.”

Response:

The erythrocyte morphology is determined by the elastic properties of erythrocyte membrane [49-51] including the contribution of shear stress deriving from membrane skeleton [52,53] and the volume to area ratio given by the relative volume v = (V2/36 pA3)1/2, where A is the membrane area and V is the enclosed volume. The relative volume is attained according to the equality of osmotic pressure inside and outside the cell [49,54] and to electro-neutrality of both compartments. Recently, additional mechanisms that are important for the erythrocyte shape such as orientational ordering of membrane constituents has been suggested [55,56]. The description of erythrocyte morphology in the introduction is upgraded (lines 112-125) and the following references (including the suggested ones) are added:

[49]        Canham, P.B. The minimum energy of bending as a possible explanation of the biconcave shape of the human red blood cell. J Theor Biol 1970, 26, 61-81, doi:10.1016/s0022-5193(70)80032-7.

[50]        Deuling, H.J.; Helfrich, W. Red blood cell shapes as explained on the basis of curvature elasticity. Biophys J 1976, 16, 861-868, doi:10.1016/s0006-3495(76)85736-0.

[51]        Martínez-Balbuena, L.; Arteaga-Jiménez, A.; Hernández-Zapata, E.; Urrutia-Buñuelos, E. Application of the Helfrich elasticity theory to the morphology of red blood cells. American Journal of Physics 2021, 89, 465-476, doi:10.1119/10.0003452.

[52]        Iglic, A. A possible mechanism determining the stability of spiculated red blood cells. J Biomech 1997, 30, 35-40, doi:10.1016/s0021-9290(96)00100-5.

[53]        Lim H W, G.; Wortis, M.; Mukhopadhyay, R. Stomatocyte-discocyte-echinocyte sequence of the human red blood cell: evidence for the bilayer- couple hypothesis from membrane mechanics. Proceedings of the National Academy of Sciences of the United States of America 2002, 99, 16766-16769, doi:10.1073/pnas.202617299.

[54] Yartsev, A. The Gibbs-Donnan effect | Deranged Physiology. Available online: https://derangedphysiology.com/main/cicm-primary-exam/required-reading/cellular-physiology/Chapter%20121/gibbs-donnan-effect (accessed on 11. 11. 2021).

[55]        Mesarec, L.; Góźdź, W.; Iglič, A.; Kralj-Iglič, V.; Virga, E.G.; Kralj, S. Normal red blood cells' shape stabilized by membrane's in-plane ordering. Sci Rep 2019, 9, 19742, doi:10.1038/s41598-019-56128-0.

[56]        Mesarec, L.; Drab, M.; Penič, S.; Kralj-Iglič, V.; Iglič, A. On the Role of Curved Membrane Nanodomains, and Passive and Active Skeleton Forces in the Determination of Cell Shape and Membrane Budding. Int J Mol Sci 2021, 22, doi:10.3390/ijms22052348.

[57]        Geekiyanage, N.; Sauret, E.; Saha, S.; Flower, R.; Gu, Y. Modelling of Red Blood Cell Morphological and Deformability Changes during In-Vitro Storage. Applied Sciences 2020, 10, doi:10.3390/app10093209.

[58]        Melzak, K.A.; Spouge, J.L.; Boecker, C.; Kirschhöfer, F.; Brenner-Weiss, G.; Bieback, K. Hemolysis Pathways during Storage of Erythrocytes and Inter-Donor Variability in Erythrocyte Morphology. Transfusion Medicine and Hemotherapy 2021, 48, 39-47, doi:10.1159/000508711.

Reviewer 2: “I would suggest the introduction to be oriented around details. Going from the vague reference to clinical applications to stored blood (aimed for transfusion) and finally to the aged cells is quite problematic. Please follow a trend and cover the relevant fields (physiology, storage conditions, time etc). If no information is available, please state so.”

Response:

We thank the reviewer for his advice on the text structuring. The introduction was rewritten (lines 41-155).

Reviewer 2: “There are two recent papers regarding transformation in association with storage and it is pretty clear that there are concerns about cell viability-functionality, e.g.

https://www.mdpi.com/2076-3417/10/9/3209

https://www.karger.com/Article/FullText/508711

Response:

We thank the reviewer for her/his suggestion of references (they were included in the lines 121-125, [57,58]). We agree that cell viability/functionality is lost under conditions yielding large vesicle production similar to the one used in our study. This is however irrelevant for a potential use of hbEVs e.g. for a test of the centrifugation efficiency.

Reviewer 2: “There are several technical questions about EVs clustering of diode discriminative ability. Please advise

Nolan, J.P. 2015. Flow cytometry of extracellular vesicles: potential, pitfalls, and prospects.Curr. Protoc. Cytom.73:13.14.1-13.14.16.doi: 10.1002/0471142956.cy1314s73 (this ref has been used in the Discussion, but the authors did not comment step by step, comparing Nolan suggestions and experimental approaches)

Edwin van der Pol et al 2018. Absolute sizing and label-free identification of extracellular vesicles by flow cytometry, Nanomedicine: Nanotechnology, Biology and Medicine. https://doi.org/10.1016/j.nano.2017.12.012

Response:

We thank the reviewer for the kind suggestion of the references. We will try to implement the approach of van der Pol et al 2018 among our FCM procedures in the future. We supplement the discussion with the lines (419-432) in which suggested references are included [87,88].

Reviewer 2: “Did the authors used any “staining”during cryomicroscopy?”

Response:

The samples were observed directly, no labelling or “staining” was introduced prior inspection.

Reviewer 2: “L344 I also don’t understand the concept of leakage. Obviously there are many questions about EVs content and transfer, but in the case of erythrocytes, it is possible that diffusion may cause the formation of these more stable structures [Marcoux, G., Duchez, AC., Cloutier, N. et al. Revealing the diversity of extracellular vesicles using high-dimensional flow cytometry analyses. Sci Rep 6, 35928 (2016). https://doi.org/10.1038/srep35928; 20 G. Fuhrmann, A. Serio, M. Mazo, R. Nair, M. M. Stevens, J. Control. Release 2015, 205, 35;  25M. Magnani, L. Rossi, M. D'ascenzo, I. Panzani, L. Bigi, A. Zanella, Biotechnol. Appl. Biochem. 1998, 28, 1]”

Response:

We have inserted this notion in the text and included the suggested references  – lines 465-466:” Clearly, many questions remain to be answered about the content, transfer-ability and diversity of EVs [89,93-95]”  , in lines 433-434: “…and powerful approaches are being developed reveal their diversity [89]”

 and lines 483-489: “In a previous study by Fuhrmann et al. [94], hypotonic dialysis was found to cause swelling and efficient loading of EVs from MDA-MB231 breast cancer cells with the hydrophilic porphyrins, caused EV aggregation, and impaired their uptake by recipient cells. The effect of hypotonic medium on the size of hbEVs was much less pronounced in our study, but as it was shown in the mentioned paper [94], that saponin was efficiently used to load EVs without altering their size. This demonstrated that integrity can be disrupted without an obvious change in EV morphology.”

[89]        Marcoux, G.; Duchez, A.-C.; Cloutier, N.; Provost, P.; Nigrovic, P.A.; Boilard, E. Revealing the diversity of extracellular vesicles using high-dimensional flow cytometry analyses. Scientific reports 2016, 6, 35928-35928, doi:10.1038/srep35928.

[93]        Glassman, P.M.; Hood, E.D.; Ferguson, L.T.; Zhao, Z.; Siegel, D.L.; Mitragotri, S.; Brenner, J.S.; Muzykantov, V.R. Red blood cells: The metamorphosis of a neglected carrier into the natural mothership for artificial nanocarriers. Advanced Drug Delivery Reviews 2021, 178, 113992, doi:https://doi.org/10.1016/j.addr.2021.113992.

[94].       Fuhrmann, G.; Serio, A.; Mazo, M.; Nair, R.; Stevens, M.M. Active loading into extracellular vesicles significantly improves the cellular uptake and photodynamic effect of porphyrins. Journal of Controlled Release 2015, 205, 35-44, doi:https://doi.org/10.1016/j.jconrel.2014.11.029.

[95]        Magnani, M.; Rossi, L.; D'Ascenzo, M.; Panzani, I.; Bigi, L.; Zanella, A. Erythrocyte engineering for drug delivery and targeting. Biotechnol Appl Biochem 1998, 28, 1-6.